# Uni-Instruct: One-step Diffusion Model through Unified Diffusion Divergence Instruction

**Yifei Wang**[1,5]   **Weimin Bai**[1,2,3]   **Colin Zhang**[4]   **Debing Zhang**[4]
**Weijian Luo**[4†]   **He Sun**[1,2,3]
[1]College of Future Technology, Peking University
[2]National Biomedical Imaging Center, Peking University
[3]Academy for Advanced Interdisciplinary Studies, Peking University
[4] hi-lab, Xiaohongshu Inc   [5] Yuanpei College, Peking University

## Abstract

In this paper, we unify more than 10 existing one-step diffusion distillation approaches, such as Diff-Instruct, DMD, SIM, SiD, $f$-distill, etc, inside a theory-driven framework which we name the ***Uni-Instruct***. Uni-Instruct is motivated by our proposed diffusion expansion theory of the $f$-divergence family. Then we introduce key theories that overcome the intractability issue of the original expanded $f$-divergence, resulting in an equivalent yet tractable loss that effectively trains one-step diffusion models by minimizing the expanded $f$-divergence family. The novel unification introduced by Uni-Instruct not only offers new theoretical contributions that help understand existing approaches from a high-level perspective but also leads to state-of-the-art one-step diffusion generation performances. On the CIFAR10 generation benchmark, Uni-Instruct achieves record-breaking Frechet Inception Distance (FID) values of ***1.46*** for unconditional generation and ***1.38*** for conditional generation. On the ImageNet $64 \times 64$ generation benchmark, Uni-Instruct achieves a new SoTA one-step generation FID of ***1.02***, which outperforms its 79-step teacher diffusion with a significant improvement margin of 1.33 (1.02 vs 2.35). We also apply Uni-Instruct on broader tasks like text-to-3D generation, which slightly outperform previous methods, such as SDS and VSD, in terms of both generation quality and diversity. Both the solid theoretical and empirical contributions of Uni-Instruct will potentially help future studies on one-step diffusion distillation and knowledge transfer of diffusion models. Code will be available at Github.

## 1 Introduction

One-step diffusion models, also known as one-step generators [28, 30], have been recognized as a stand-alone family of generative models that reach the leading generative performances in a wide range of applications, including benchmarking image generation [30, 31, 62, 69, 68, 17, 59], text-to-image generation [29, 32, 61, 62, 31, 67, 16], text-to-video generation [1, 63], image-editing [13], and numerous others [34, 41, 55].

Currently, the mainstream of training the one-step diffusion model is through proper distillation approaches that minimize divergences between distributions of the one-step model and some teacher diffusion models. For instance, Diff-Instruct[30] was the first work that introduced one-step diffusion models by minimizing the Kullback-Leibler divergence. DMD [62] improves the Diff-Instruct by introducing an additional regression loss. Score-identity Distillation (SiD) [69] studies the one-step diffusion distillation by minimizing the Fisher divergence, but without the proof of gradient

---

†Correspondence to `pkulwj1994@icloud.com`.

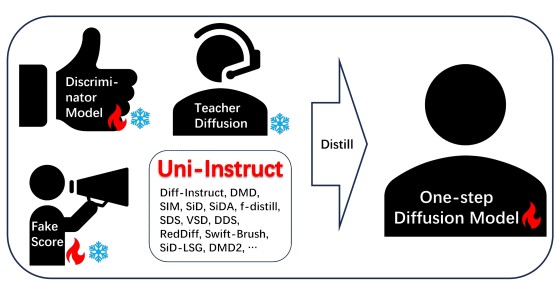 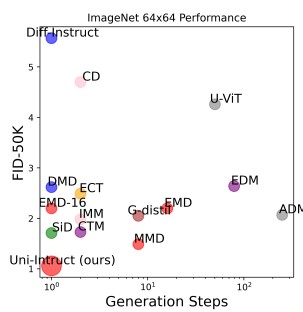

Figure 1: **Left:** Conception overview of *Uni-Instruct*. The *Uni-Instruct* unifies more than 10 existing diffusion distillation methods in a wide range of applications. Please check Table 1 for details. **Right:** selected FID scores of different models on ImageNet-$64 \times 64$ conditional generation benchmark.

equivalence of the loss function. Later, Score Implicit Matching (SIM)[31] introduced a complete proof of the gradient equivalence of losses that minimizes the general score-based divergence family, including the Fisher divergence as a special case. f-distill [59] and SiDA [68] recently generalized the Diff-Instruct and the SiD to the integral $f$-divergence and auxiliary GAN losses, resulting in performance improvements on image generation benchmarks. Other approaches have also elaborated on the one-step diffusion models on a wide range of applications through the lens of divergence minimization [16, 68, 67, 61, 29, 32, 17].

Provided that existing one-step diffusion models have achieved impressive performances, with some of them even outperforming their multi-step teacher diffusions, existing training approaches can be conceptually separated into two lines:

**(1)** Diff-Instruct[30], and its variants like DMD[62], tackle the Integral Kullback-Leibler Divergence, while f-distill[59] unifies the IKL as a special case of Integral f-divergence. These KL and $f$-divergence based distillation approaches have the advantage of fast convergence, but suffer from mode-collapse issues and sub-optimal performances;

**(2)** Score Implicit Matching (SIM[31]) proves a solid theoretical equivalence of score-based divergence minimization, which unifies the SiD[69] and Fisher divergences as special cases. Though these general score-based divergences minimization has shown surprising generation performance, they may suffer from slow convergence issues and sub-optimal fidelity.

Till now, it seems that the KL-based and Score-based divergence minimization approaches are pretty parallel in theory. Therefore, we are strongly motivated to answer an interesting yet important research question: ***Can we unify KL-based and Score-based approaches in a unified theoretical framework? If we can, would the unified approach lead to better one-step diffusion models?***

In this paper, we provide a complete answer to the mentioned question. We successfully built a unified theoretical framework based on a novel diffusion expansion of the $f$-divergence family. Though the original expanded $f$-divergence family is not tractable to optimize, we introduced new theorems that lead to tractable yet equivalent losses, therefore making Uni-Instruct an executable training method.

In this way, we are able to unify more than 10 existing diffusion distillation methods across a wide range of applications via our proposed ***Uni-Instruct***. The methods that have been unified by Uni-Instruct include both KL-divergence based methods (such as Diff-Instruct[30], DMD[62], and $f$-distill[59]) and general score-divergence based methods (such as Score Implicit Matching (SIM[31]), SiD[69], and SiDA[68]), as is shown in Table 1. Such a novel unification of existing one-step diffusion models marks the uniqueness of Uni-Instruct, which brings new perspectives in understanding and connecting different one-step diffusion models. Besides the solid theoretical contributions, Uni-Instruct also leads to new State-of-the-art one-step image generation performances on competitive image generation benchmarks: it achieved a record-breaking FID (Fréchet Inception Distance) [14] value of ***1.02*** on the ImageNet $64 \times 64$ conditional generation task. This score outperforms its 79-step teacher diffusion with a significant improvement margin of 1.33 (1.02 vs 2.35). Uni-Instruct also leads to new state-of-the-art one-step FIDs of ***1.46*** on CIFAR10 conditional generation task and ***1.36*** on CIFAR10 unconditional generation task, significantly outperforming previous one-step models such as f-distill, SiDA, SIM, SiD, DMD, and Diff-Instruct. It also outperforms competitive few-step

| Method | Loss | Div. in UI | Task | Loss Function | Gradient Expression |
|---|---|---|---|---|---|
| Diff-Instruct (DI) [30] | IKL Div. | $\chi^2$ | One-Step Diffusion | $\int w(t)\mathcal{D}_{\text{KL}}(p_{\theta,t}||q_0)\mathrm{d}t$ | $(\boldsymbol{s}_{p_{\theta,t}}(\boldsymbol{x}_t) - \boldsymbol{s}_{q_t}(\boldsymbol{x}_t))\frac{\partial \mathbf{x}_t}{\partial \theta}$ |
| DI++ [29] | IKL Div. + Reward | $\chi^2$ | Human Aligned One-Step Diffusion | $\int w(t)\mathcal{D}_{\text{KL}}(p_{\theta,t}||q_0)\mathrm{d}t$ $+\mathcal{L}_{\text{reward}}$ | $\text{Grad(DI)} + \nabla_\theta \mathcal{L}_{\text{reward}}$ |
| DI* [32] | KL Div. + Reward | RKL | Human Aligned One-Step Diffusion | $\mathcal{D}_{\text{KL}}(p_{\theta,t}||q_0)$ $+\mathcal{L}_{\text{reward}}$ | $\text{Grad(SIM)} + \nabla_\theta \mathcal{L}_{\text{reward}}$ |
| SDS [41] | IKL Div. | $\chi^2$ | Text to 3D | $\int w(t)\mathcal{D}_{\text{KL}}(p_{\theta,t}||q_0)\mathrm{d}t$ | $\text{Grad(DI)}$ |
| DDS [13] | IKL Div. | $\chi^2$ | Image Editing | $\int w(t)\mathcal{D}_{\text{KL}}(p_{\theta,t}||q_0)\mathrm{d}t$ | $\text{Grad(DI)}$ |
| VSD [55] | IKL Div. | $\chi^2$ | Text to 3D | $\int w(t)\mathcal{D}_{\text{KL}}(p_{\theta,t}||q_0)\mathrm{d}t$ | $\text{Grad(DI)}$ |
| DMD [62] | IKL Div. + Reg. | $\chi^2$ | One-Step Diffusion | $\int w(t)\mathcal{D}_{\text{KL}}(p_{\theta,t}||q_0)\mathrm{d}t$ $+\mathcal{L}_{\text{MSE}}$ | $\text{Grad(DI)} + \nabla_\theta \text{MSE}$ |
| RedDiff [34] | IKL Div. + Data Fedility | $\chi^2$ | Inverse Problem | $\int w(t)\mathcal{D}_{\text{KL}}(p_{\theta,t}||q_0)\mathrm{d}t$ $+\mathcal{L}_{\text{MSE}}$ | $\text{Grad(DI)} + \nabla_\theta \text{MSE}$ |
| DMD2 [61] | IKL Div. + GAN | $\chi^2$ | One-Step Diffusion | $\int w(t)\mathcal{D}_{\text{KL}}(p_{\theta,t}||q_0)\mathrm{d}t$ $+\mathcal{L}_{adv.}$ | $\text{Grad(DI)} + \nabla_\theta \mathcal{L}_{adv.}$ |
| Swift Brush [16] | IKL Div. | $\chi^2$ | One-Step Diffusion | $\int w(t)\mathcal{D}_{\text{KL}}(p_{\theta,t}||q_0)\mathrm{d}t$ | $\text{Grad(DI)}$ |
| SIM [31] | General KL Div. | RKL | One-Step Diffusion | $\mathcal{D}_{\text{KL}}(p_{\theta,t}||q_0)$ | $\frac{\partial}{\partial\theta}\left(\boldsymbol{s}_{q_t}(\boldsymbol{x}_t) - \boldsymbol{s}_{p_{\theta,t}}(\boldsymbol{x}_t)\right)\cdot$ $\left(\boldsymbol{s}_{P_{sg[\theta]},t}(\boldsymbol{x}_t) - \nabla \log q_t(\boldsymbol{x}_t|\boldsymbol{x}_0)\right)$ |
| SiD [69] | KL Div. | RKL | One-Step Diffusion | $\mathcal{D}_{\text{KL}}(p_{\theta,t}||q_0)$ | $\text{Grad(SIM)}$ |
| SiDA [68] | KL Div. + GAN | RKL | One-Step Diffusion | $\mathcal{D}_{\text{KL}}(p_{\theta,t}||q_0)+\mathcal{L}_{adv.}$ | $\text{Grad(SIM)} + \nabla_\theta \mathcal{L}_{adv.}$ |
| SiD-LSG [67] | KL Div. | RKL | One-Step Diffusion | $\mathcal{D}_{\text{KL}}(p_{\theta,t}||q_0)$ | $\text{Grad(SIM)}$ |
| $f$-distill [59] | I-$f$ Div. + GAN | $\chi^2$ | One-Step Diffusion | $\int w(t)\mathcal{D}_f(q_0||p_{\theta,t})\mathrm{d}t$ $+\mathcal{L}_{adv.}$ | $\lambda_f\text{Grad(DI)} + \nabla_\theta \mathcal{L}_{adv.}$ |
| Uni-Instruct **(Ours)** | $f$ Div. + GAN | **All** | **All** | $\mathcal{D}_f(q_0||p_{\theta,t})$ $+\mathcal{L}_{adv.}$ | $\nabla_\theta \mathcal{L}_{adv.} + \lambda_f^{\text{DI}}\text{Grad(DI)}$ $+\lambda_f^{\text{SIM}}(\mathbf{x})\text{Grad(SIM)}$ |

Table 1: Distribution matching diffusion distillation loss family. Uni-Instruct not only extends the distribution matching framework theoretically, but also unifies all previous gradient expressions with specific weightings.

generative models, including consistency models [11, 48, 25], moment matching distillation models [44], inductive models [66], and many others [57].

Besides the one-step generation benchmark, we are inspired by DreamFusion [41], ProlificDreamer [55], SIM [31]'s application on 3D experiments. In Section 4.3, we also successfully apply Uni-Instruct as a knowledge transferring approach for text-to-3D generation applications, resulting in robust, diverse, and high-fidelity 3D contents which are slightly better than ProlificDreamer in quality and diversity.

We summarize the theoretical and practical contributions in this paper as follows:

- **Unified Theoretical Framework:** We introduced a unified theoretical framework named *Uni-Instruct* together with a novel $f$-divergence expansion theorem. *Uni-Instruct* is able to unify more than 10 existing one-step diffusion distillation approaches, bringing new perspectives to understanding one-step diffusion models.

- **Tractable and Flexible Training Objective:** We introduce novel theoretical tools, such as gradient equivalence theorems, and derived tractable yet equivalent losses for Uni-Instruct. This leads to both flexible training objectives and new tools for one-step diffusion models.

- **New SoTA Practical Performances:** Uni-Instruct achieved new state-of-the-art generation performances (measured in FID) on CIFAR10 (a one-step FID of 1.36) and ImageNet$64 \times 64$ (a one-step FID of 1.02) benchmarks. We also successfully applied Uni-Instruct on the text-to-3D generation task, resulting in plausible and diverse 3D generation results.

## 2 Preliminary

### 2.1 Diffusion Models

**Diffusion Models.** Assume we observe data from the underlying distribution $q_d(\boldsymbol{x})$. The goal of generative modeling is to train models to generate new samples $\boldsymbol{x} \sim q_d(\boldsymbol{x})$. The forward diffusion

process of DM transforms any initial distribution $q_0 = q_d$ towards some simple noise distribution,

$$\mathrm{d}\boldsymbol{x}_t = \boldsymbol{F}(\boldsymbol{x}_t, t)\mathrm{d}t + g(t)\mathrm{d}\boldsymbol{w}_t, \tag{2.1}$$

where $\boldsymbol{F}$ is a pre-defined drift function, $g(t)$ is a pre-defined scalar-value diffusion coefficient, and $\boldsymbol{w}_t$ denotes an independent Wiener process. A continuous-indexed score network $\boldsymbol{s}_\varphi(\boldsymbol{x}, t)$ is employed to approximate marginal score functions of the forward diffusion process (2.1). The learning of score networks is achieved by minimizing a weighted denoising score matching objective [53, 50],

$$\mathcal{L}_{\mathrm{DSM}}(\varphi) = \int_{t=0}^{T} \lambda(t)\mathbb{E}_{\boldsymbol{x}_0 \sim q_0, \boldsymbol{x}_t|\boldsymbol{x}_0 \sim q_{t|0}(\boldsymbol{x}_t|\boldsymbol{x}_0)}\|\boldsymbol{s}_\varphi(\boldsymbol{x}_t, t) - \nabla_{\boldsymbol{x}_t} \log q_t(\boldsymbol{x}_t|\boldsymbol{x}_0)\|_2^2 \mathrm{d}t. \tag{2.2}$$

Here, the weighting function $\lambda(t)$ controls the importance of the learning at different time levels, and $q_t(\boldsymbol{x}_t|\boldsymbol{x}_0)$ denotes the conditional transition of the forward diffusion (2.1). After training, the score network $\boldsymbol{s}_\varphi(\boldsymbol{x}_t, t) \approx \nabla_{\boldsymbol{x}_t} \log q_t(\boldsymbol{x}_t)$ is a good approximation of the marginal score function of the diffused data distribution. High-quality samples from a DM can be drawn by simulating SDE, which is implemented by the learned score network [50]. However, the simulation of an SDE is significantly slower than that of other models, such as one-step generator models.

## 2.2 One-step Diffusion Model via KL Divergence Minimization

**Notations and the Settings of One-step Diffusion Models.** We use the traditional settings introduced in Diff-Instruct [30] to present one-step diffusion models. Our basic setting is that we have a pre-trained diffusion model specified by the score function $\boldsymbol{s}_{q_t}(\boldsymbol{x}_t) := \nabla_{\boldsymbol{x}_t} \log q_t(\boldsymbol{x}_t)$ where $q_t(\boldsymbol{x}_t)$'s are the underlying distribution diffused at time $t$ according to (2.1). We assume that the pre-trained diffusion model provides a sufficiently good approximation of the data distribution, and thus will be the only item of consideration for our approach.

The one-step diffusion model of our interest is a single-step generator network $g_\theta$, which can transform an initial random noise $\boldsymbol{z} \sim p_z$ to obtain a sample $\boldsymbol{x} = g_\theta(\boldsymbol{z})$; this network is parameterized by network parameters $\theta$. Let $p_{\theta,0}$ denote the data distribution of the student model, and $p_{\theta,t}$ denote the marginal diffused data distribution of the student model with the same diffusion process (2.1). The student distribution implicitly induces a score function $\boldsymbol{s}_{p_{\theta,t}}(\boldsymbol{x}_t) := \nabla_{\boldsymbol{x}_t} \log p_{\theta,t}(\boldsymbol{x}_t)$, and evaluating it is generally performed by training an alternative score network as elaborated later.

**Diff-Instruct (DI).** Diff-Instruct [30] is the first work that trains one-step diffusion models by minimizing the integral of KL divergence between the one-step model and the teacher diffusion model distributions. The integral Kullback-Leibler divergence between one-step model $p_\theta(.)$ and teacher diffusion model $q_0(.)$ is defined as: $\mathcal{D}_{\mathrm{IKL}}(p_\theta\|q_0) := \int_{t=0}^{T} w(t)\mathbb{E}_{\substack{\boldsymbol{x}_0 = g_\theta(\boldsymbol{z}),\ \boldsymbol{z} \sim \mathcal{N}(\boldsymbol{0}, \boldsymbol{I}) \\ \boldsymbol{x}_t|\boldsymbol{x}_0 \sim q_{t|0}(\boldsymbol{x}_t|\boldsymbol{x}_0)}} \left\{ \log \frac{p_{\theta,t}(\boldsymbol{x}_t)}{q_t(\boldsymbol{x}_t)} \right\}\mathrm{d}t$.

Though IKL as a training objective is intractable because we do not have a direct dependence of $\theta$ and $p_{\theta,t}(.)$. [30] proved in theory that a tractable yet equivalent objective writes:

$$\mathcal{L}_{\mathrm{DI}}(\theta) := \int_{t=0}^{T} w(t)\mathbb{E}_{\substack{\boldsymbol{x}_0 = g_\theta(\boldsymbol{z}),\ \boldsymbol{z} \sim \mathcal{N}(\boldsymbol{0}, \boldsymbol{I}) \\ \boldsymbol{x}_t|\boldsymbol{x}_0 \sim q_{t|0}(\boldsymbol{x}_t|\boldsymbol{x}_0)}} \mathrm{SG}\left\{ \boldsymbol{s}_{p_{\mathrm{SG}[\theta]},t}(\boldsymbol{x}_t) - \boldsymbol{s}_{q_t}(\boldsymbol{x}_t) \right\}^T \boldsymbol{x}_t(\theta)\mathrm{d}t, \tag{2.3}$$

Where the operator $\mathrm{SG}(\cdot)$ in (2.3) represents the stop-gradient operator. Diff-Instruct proposed to use an online-trained fake diffusion model to approximate the stopped-gradient one-step model score function $\boldsymbol{s}_{\psi,t}(\boldsymbol{x}_t) \approx \boldsymbol{s}_{p_{\mathrm{SG}[\theta]},t}(\boldsymbol{x}_t)$. Such a novel use of a fake score is kept by following approaches such as DMD, SiD, etc. Two key contributions of Diff-Instruct are (1) first introducing the concept of the one-step distillation via divergence minimization; (2) introducing a technical path that derives tractable losses by proving gradient equality w.r.t the intractable divergence.

## 2.3 One-step Diffusion Model via Score-based Divergence Minimization

**Score Implicit Matching (SIM).** Inspired by Diff-Instruct and the empirical success of SiD [69], recent work, the Score-implicit Matching (SIM) [31], has generalized the KL divergences to general score-based divergence by proving new gradient equivalence theories. The general score-divergence is defined via: $\mathcal{D}^{[0,T]}(p, q) := \int_{t=0}^{T} w(t)\mathbb{E}_{\boldsymbol{x}_t \sim \pi_t}\left\{ \mathbf{d}(\boldsymbol{s}_{p_t}(\boldsymbol{x}_t) - \boldsymbol{s}_{q_t}(\boldsymbol{x}_t)) \right\}\mathrm{d}t$, where $p_t$ and $q_t$ denote the marginal densities of the diffusion process (2.1) at time $t$ initialized with $q$ and $p$ respectively.

$w(t)$ is an integral weighting function. $\mathbf{d}(\cdot)$ is a distance function. Clearly, we have $\mathcal{D}^{[0,T]}(p,q) = 0$ if and only if all marginal score functions agree, which implies that $p_0(\boldsymbol{x}_t) = q_0(\boldsymbol{x}_t)$, $a.s.$ $\pi_0$.

SIM shows that Eq. (2.4) has the same parameter gradient as the intractable score-divergence:

$$\mathcal{L}_{\text{SIM}}(\theta) = \int_{t=0}^{T} w(t)\mathbb{E}_{\substack{\boldsymbol{z} \sim p_z, \boldsymbol{x}_0 = g_\theta(\boldsymbol{z}), \\ \boldsymbol{x}_t | \boldsymbol{x}_0 \sim q_t(\boldsymbol{x}_t | \boldsymbol{x}_0)}} \left\{ -\mathbf{d}'(\boldsymbol{y}_t) \right\}^T \left\{ \boldsymbol{s}_{p_{\text{sg}[\theta],t}}(\boldsymbol{x}_t) - \nabla_{\boldsymbol{x}_t} \log q_t(\boldsymbol{x}_t | \boldsymbol{x}_0) \right\} \mathrm{d}t, \quad (2.4)$$

with $\boldsymbol{y}_t := \boldsymbol{s}_{p_{\text{sg}[\theta],t}}(\boldsymbol{x}_t) - \boldsymbol{s}_{q_t}(\boldsymbol{x}_t)$. Now the objective becomes tractable.

In Section 3, we use theoretical tools from Diff-Instruct and SIM to prove the gradient equivalence of tractable Uni-Instruct loss and the intractable expanded $f$-divergence. Furthermore, we are surprisingly to find that the resulting gradient expression recovers a novel combination of the Diff-Instruct and the SIM parameter gradient.

### 2.4 Relation Between KL Divergence and Fisher Divergence

Inspired by the famous De Bruijn identity [60, 6] that describes entropy evolution along heat diffusion, notable works [45, 37, 33, 49] have built the relationship between KL divergence and Fisher divergence via a diffusion expansion: the KL divergence is the integral of the Fisher divergence along a diffusion process under mild regularity conditions:

$$\mathcal{D}_{\text{KL}}(p_\theta || q_0) = \int_0^T \frac{1}{2} g^2(t) \mathbb{E}_{p_\theta} \left[ ||\boldsymbol{s}_{p_t}(\boldsymbol{x}_t) - \boldsymbol{s}_{q_t}(\boldsymbol{x}_t)||_2^2 \right] dt \quad (2.5)$$

Motivated by the relationship between KL divergence and Fisher divergence, in Section 3, we begin the Uni-Instruct framework by proposing a novel diffusion expansion theorem of general KL divergence: the $f$-divergence family.

## 3 Uni-Instruct: Unify One-step Distillation Methods in Theory

In this section, we introduce Uni-Instruct, a theory-driven family of approaches for the one-step distillation of score-based diffusion models. Uni-Instruct is able to unify more than 10 existing methods as special cases with proper weighting functions. It also leads to new state-of-the-art one-step generation performances on ImageNet$64 \times 64$ and CIFAR10 generation benchmarks.

Uni-Instruct is built upon a novel *diffusion expansion* theory of the $f$-divergence family. We begin by giving a brief introduction to the $f$-divergence family. We then prove a novel diffusion expansion theory of $f$-divergences in Section 3.1, which acts as the target objective we would like to optimize. Then in Section 3.2, we provide a non-trivial theorem that leads to an equivalent yet tractable loss function that shares the same parameter gradient as the intractable expanded $f$-divergence.

### 3.1 Diffusion Expansion of $f$-Divergence

$f$-**divergence.** For a convex function $f(\cdot)$ on $(0, +\infty)$, where $f(1) = 0$, The $f$-divergence[42] is:

$$\mathcal{D}_f(q||p) = \int p(\mathbf{x}) f\left(\frac{q(\mathbf{x})}{p(\mathbf{x})}\right) d\mathbf{x}. \quad (3.1)$$

Appropriate choices of the function $f(\cdot)$ lead to many widely-used divergences such as reverse-KL divergence (RKL), forward-KL divergence (FKL), Jeffrey-KL divergence (JKL), Jensen-Shannon divergence (JS), and Chi-Square divergence ($\chi^2$). We put more introductions in the appendix C.

**The Diffusion Expansion Theorem.** We use the same notations and settings in Section 2.2. $g_\theta(\cdot)$ represents the one-step diffusion model, and $q_t(\cdot)$ represents the distributions of the teacher diffusion model. Our goal is to minimize the $f$-divergence between the output image distribution of the one-step model's distribution and the teacher diffusion model distribution $D_f(q_0 || p_\theta)$. However, since $f$-divergences are defined in the image data space, they can not directly incorporate instructions from multiple noise levels of teacher diffusion models. To address this issue, we first introduce a diffusion expansion Theorem 3.1 of $f$-divergence along a diffusion process. This expansion enables us to construct training objectives by considering all diffusion noise levels.

**Theorem 3.1** (Diffusion Expansion of $f$-Divergence). *Assume $p, q$ are distributions that both evolve along Eq. 2.1. We have the following equivalence:*

$$\mathcal{D}_f(q_0\|p_\theta) = \int_0^T \frac{1}{2}g^2(t)\mathbb{E}_{p_{\theta,t}}\left[\left(\frac{q_t}{p_{\theta,t}}\right)^2 f''\left(\frac{q_t}{p_{\theta,t}}\right)\|\boldsymbol{s}_{p_{\theta,t}}(\boldsymbol{x}_t) - \boldsymbol{s}_{q_t}(\boldsymbol{x}_t)\|_2^2\right]\mathrm{d}t + \mathcal{D}_f(q_T\|p_{\theta,T}),$$

(3.2)

We give a complete proof with regularity analysis in Appendix B.1. For simplicity, we assume $\mathcal{D}_f(q_T\|p_{\theta,T}) = 0$ and ignore the last term in Eq. 3.2 in the following section. This fundamental expansion (Eq. 3.2) expands the static $f$-divergence in data space into an integral of divergences along the diffusion process. ***However, directly optimizing objective*** (3.2) ***is not tractable*** because we do not know the exact expressions of either the density $p_{\theta,t}$ or the score function $\boldsymbol{s}_{p_{\theta,t}}(\cdot)$ of the diffused one-step model's distribution. To step towards a tractable objective, we derive the $\theta$ gradient of the expanded $f$-divergence (3.2) in Theorem 3.2.

## 3.2 Theories to Get Tractable Losses

To tackle the intractable issue of the expanded $f$-divergence, we prove a novel parameter gradient equivalence theorem 3.2.

**Theorem 3.2** (Gradient Equality Theorem of the Expanded $f$-divergence). *Let $q_t(\boldsymbol{x})$ and $p_{\theta,t}(\boldsymbol{x})$ be probability density functions evolving under the Fokker-Planck dynamics, and $f : \mathbb{R}_+ \to \mathbb{R}$ is a four-times differentiable convex function. The parameter gradient of the $f$-divergence rate satisfies:*

$$\frac{1}{2}g^2(t)\nabla_\theta\left\{\mathbb{E}_{p_{\theta,t}}\left[\left(\frac{q_t}{p_{\theta,t}}\right)^2 f''\left(\frac{q_t}{p_{\theta,t}}\right)\|\boldsymbol{s}_{p_{\theta,t}}(\boldsymbol{x}_t) - \boldsymbol{s}_{q_t}(\boldsymbol{x}_t)\|_2^2\right]\right\}$$

$$= -\frac{1}{2}g^2(t)\frac{\partial}{\partial\theta}\left\{\mathbb{E}_{p_{\theta,t}}\left[SG\left(\mathcal{C}_1\left(\frac{q_t}{p_{\theta,t}}\right)\right)2\left(\boldsymbol{s}_{q_t}(\boldsymbol{x}_t) - \boldsymbol{s}_{p_{sg[\theta],t}}(\boldsymbol{x}_t)\right)\left(\boldsymbol{s}_{p_{sg[\theta],t}}(\boldsymbol{x}_t) - \nabla_{\boldsymbol{x}_t}\log q_t(\boldsymbol{x}_t \mid \boldsymbol{x}_0)\right)\right]\right\}$$

$$- \frac{1}{2}g^2(t)\frac{\partial}{\partial\theta}\left\{\mathbb{E}_{p_{\theta,t}}\left[SG\left(\mathcal{C}_2\left(\frac{q_t}{p_{\theta,t}}\right)\left(\boldsymbol{s}_{q_t}(\boldsymbol{x}_t) - \boldsymbol{s}_{p_{\theta,t}}(\boldsymbol{x}_t)\right)\|\boldsymbol{s}_{q_t}(\boldsymbol{x}_t) - \boldsymbol{s}_{p_{\theta,t}}(\boldsymbol{x}_t)\|_2^2\right)\boldsymbol{x}_t\right]\right\}$$

(3.3)

*where SG donates stop gradient operator, and the curvature coupling coefficient $\mathcal{C}(r)$ are defined as:*

$$\mathcal{C}_1(r) := r^3 f'''(r), \quad \mathcal{C}_2(r) := 2r^2 f''(r) + 4r^3 f'''(r) + r^4 f''''(r), \quad r := \frac{q_t(\boldsymbol{x})}{p_{\theta,t}(\boldsymbol{x})}$$

(3.4)

**Remark 3.3.** It is worth noting that in Theorem 3.2, we derived an equality of the gradient of the intractable expanded $f$-divergence. The right side of the equality is two terms, which are gradients of two tractable functions. With this observation, we can see that minimizing the tractable right-hand side of equality (3.3) using gradient-based optimization algorithms such as Adam [21] is equivalent to minimizing the intractable expanded $f$-divergence, which lies in the left-hand side.

We notice that the gradient of the training objective admits a composition of the Diff-Instruct[30] gradient and a SIM [31] gradient. Therefore, we can formally write down our tractable loss function as:

$$\mathcal{L}_{UI}(\theta) = \int_0^T -\frac{1}{2}g^2(t)\left(\lambda_f^{\mathrm{DI}}\mathcal{L}_{\mathrm{DI}} + \lambda_f^{\mathrm{SIM}}\mathcal{L}_{\mathrm{SIM}}\right)\mathrm{d}t,$$

(3.5)

$$\mathcal{L}_{\mathrm{SIM}} = \mathbb{E}_{p_{\theta,t}}\left[SG\left(\mathcal{C}_1\left(r\right)\right)2\left(\boldsymbol{s}_{q_t}(\boldsymbol{x}_t) - \boldsymbol{s}_{p_{sg[\theta],t}}(\boldsymbol{x}_t)\right)\left(\boldsymbol{s}_{p_{sg[\theta],t}}(\boldsymbol{x}_t) - \nabla_{\boldsymbol{x}_t}\log q_t(\boldsymbol{x}_t \mid \boldsymbol{x}_0)\right)\right],$$

$$\mathcal{L}_{\mathrm{DI}} = \mathbb{E}_{p_{\theta,t}}\left[SG\left(\mathcal{C}_2\left(r\right)\left(\boldsymbol{s}_{q_t}(\boldsymbol{x}_t) - \boldsymbol{s}_{p_{\theta,t}}(\boldsymbol{x}_t)\right)\|\boldsymbol{s}_{q_t}(\boldsymbol{x}_t) - \boldsymbol{s}_{p_{\theta,t}}(\boldsymbol{x}_t)\|_2^2\right)\boldsymbol{x}_t\right],$$

(3.6)

where the weighting coefficients are determined by the $f$-divergence selection, we provide our completed proofs in Appendix B.2.

**Density Ratio Estimation via an Auxiliary GAN Loss**    Notice that the tractable loss function (3.5) requires the density ratio between the one-step model and teacher diffusion. For this, we train a GAN discriminator along the process, where the discriminator output serves as an estimator. This use of GAN discriminator is also widely applicable in other works like SiDA[68] and $f$-distill [59]. Details on why the GAN discriminator recovers the density ratio can be found in Theorem B.1.

**Practical Algorithm of the Uni-Instruct**   We can now present the formal training algorithm of Uni-Instruct. As is shown in Algorithm 1, we maintain the active training status of three models: one-step diffusion model, online fake score network, and a discriminator. The training is performed in two steps alternatively: we first optimize the discriminator with real data, and then optimize the online fake score network with score matching loss. After that, we optimize the one with Uni-Instruct loss, which is given by the previous two models. Uni-Instruct loss varies based on the divergence we choose. We provide example divergences in Tab. 8. Note that through choosing proper divergence, we can recover the distillation loss of Diff-Instruct [30], SIM [31], as well as $f$-distill [59]. To be more specific: $\mathcal{L}_{\text{SIM}}$ vanishes when selecting $\chi^2$-divergence, while $\mathcal{L}_{\text{DI}}$ vanishes if we choose forward-KL, reverse-KL, and Jeffrey-KL divergence.

---

**Algorithm 1:** Uni-Instruct Algorithm on Distilling One Step Diffusion Model

---

**Input:** pre-trained DM $s_{q_t}$, generator $g_\theta$, fake score network $s_\psi$, discriminator $D_\lambda$, divergence $f$, GAN weight $w_{\text{GAN}}$, diffusion timesteps weighting $w(t)$.

1: **while** not converge **do**
2:    Sample real images and random noises: $\boldsymbol{x}_{\text{real}} \sim p_{\text{data}}, \epsilon \sim \mathcal{N}(0, I)$
3:    Generate fake images: $\boldsymbol{x}_{\text{fake}} = g_\theta(\epsilon)$
4:    Update $D_\lambda$ with discriminator loss:
     $\mathcal{L}_D = -\mathbb{E}_{\boldsymbol{x}_{\text{real}}}[\log D_\lambda(\boldsymbol{x}_{\text{real}})] - \mathbb{E}_{\boldsymbol{x}_{\text{fake}}}[\log(1 - D_\lambda(\boldsymbol{x}_{\text{fake}}))]$
5:    Update $s_\psi$ with denoising score matching loss:
     $\mathcal{L}_{\text{diffusion}} = \int_0^T w(t) \, \mathbb{E}_{\boldsymbol{x}_t | \boldsymbol{x}_{\text{fake}} \sim p_{\theta,t}(\boldsymbol{x}_t | \boldsymbol{x}_{\text{fake}})} \| s_\psi(\boldsymbol{x}_t, t) - \nabla_{\boldsymbol{x}_t} \log p_t(\boldsymbol{x}_t \mid \boldsymbol{x}_{\text{fake}}) \|_2^2 \, \mathrm{d}t$
6:    Calculate Uni-Instruct loss: $\mathcal{L}_{\text{Uni}} = $ Equation 3.5
7:    Calculate adversarial loss (non-saturating): $\mathcal{L}_{\text{GAN}} = -\mathbb{E}_{\boldsymbol{x}_{\text{fake}}}[\log D_\lambda(\boldsymbol{x}_{\text{fake}})]$
8:    Update $g_\theta$ with total loss: $\mathcal{L}_{\text{total}} = \mathcal{L}_{\text{Uni}} + w_{\text{GAN}} \cdot \mathcal{L}_{\text{GAN}}$
9: **end while**
10: **return** $g_\theta$

---

## 3.3   How Uni-Instruct can Unify Previous Methods

In this section, we show in what cases Uni-Instruct can recover previous methods. As is shown in Tab. 1, Uni-Instruct can effectively unify more than 10 existing distillation methods for one-step diffusion models, such as Diff-Instruct, DMD, $f$-distill, SIM, and SiD.

**DI, DMD, and $f$-distill are Uni-Instruct with additional time weighting.**   DI [30] and DMD [62] integrates KL divergence along a diffusion process: $D_{\text{IKL}}(p_\theta || q_0) := \int_0^T w(t) D_{\text{KL}}(p_\theta || q_0) dt$. Furthermore, $f$-distill [59] replace KL with general $f$-divergence. Our goal, on the other hand, is to match these two distributions only at the original distributions: $D_f(q_0 || p_\theta)$, which requires no specific weightings $\omega(t)$. Our framework is more theoretically self-consistent for those ad-hoc weightings that may induce mismatches between the optimization target and the true distribution divergence. However, with additional weightings, Uni-Instruct can recover $f$-distill.

**Corollary 3.4.** *Suppose $W(t) = \int w(t)dt + C, W(0) = 0$, the expression of Uni-Instruct with an extra weighting $W(t)$ is equivalent to $f$-distill:*

$$\int_0^T \frac{1}{2} g^2(t) W(t) \mathbb{E}_{p_{\theta,t}} \left[ \left( \frac{q_t}{p_{\theta,t}} \right)^2 f'' \left( \frac{q_t}{p_{\theta,t}} \right) \| s_{p_{\theta,t}}(\boldsymbol{x}_t) - s_{q_t}(\boldsymbol{x}_t) \|_2^2 \right] \mathrm{d}t = \int_0^T w(t) \mathcal{D}_f(q_0 || p_{\theta,t}) \mathrm{d}t. \quad (3.7)$$

Complete proof is in Appendix B.4, which leverages integration by parts and Theorem 3.1.

**SIM is a Special Case of Uni-Instruct.**   Suppose $\mathbf{d}(\cdot)$ is l2-norm, SIM in Section 2.3 becomes: $\int_0^T \omega(t) \mathbb{E}_{p_{\theta,t}} \left[ \| s_{p_{\theta,t}}(\boldsymbol{x}_t) - s_{q_t}(\boldsymbol{x}_t) \|_2^2 \right] \mathrm{d}t$. It turns out that SIM is a special case of Uni-Instruct. We find that the right-hand side of Theorem 3.1 will degenerate to SIM through selecting the divergence as reverse-KL divergence: $\mathcal{D}_{\text{KL}}(p_\theta || q_0) = \frac{1}{2} \int_0^T g^2(t) \mathbb{E}_{p_{\theta,t}} \left[ \| s_{p_{\theta,t}}(\boldsymbol{x}_t) - s_{q_t}(\boldsymbol{x}_t) \|_2^2 \right] \mathrm{d}t$. As a result, SIM is secretly minimizing the KL divergence between the teacher model and the one-step diffusion model, which is a special case of our $f$-divergence. Beyond this specific configuration, Uni-Instruct offers enhanced flexibility through its support for alternative divergence metrics, including FKL and JKL, which enable improved mode coverage. This generalized formulation contributes to superior empirical performance, achieving lower FID values.

## 3.4 Text-to-3D Generation using Uni-Instruct

Recent advances in 3D text-to-image synthesis leverage 2D diffusion models as priors. Dreamfusion [41] introduced score distillation sampling (SDS) to align NeRFs with text guidance, while Prolific-Dreamer [55] improved quality via variational score distillation (VSD). These methods mainly use reverse KL divergence. Uni-Instruct generalizes this framework by allowing flexible divergence choices (e.g., FKL, JKL), enhancing mode coverage and geometric fidelity, and unifying SDS and VSD as special cases.

## 4 Experiments

In this section, we first demonstrate Uni-Instruct's strong capability to generate high-quality samples on benchmark datasets through efficient distillation. Followed by text-to-3D generation, which illustrates the wide application of Uni-Instruct.

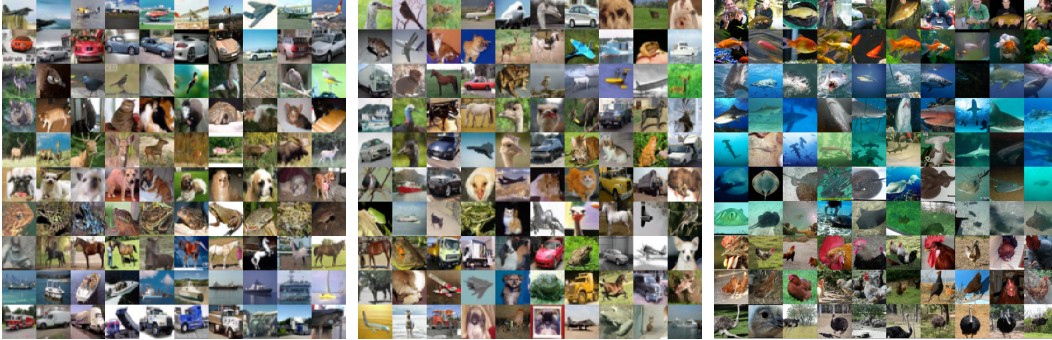

Figure 2: Generated samples from Uni-Instruct one-step generators that are distilled from pre-trained diffusion models on different datasets. *Left:* CIFAR10 (unconditional); *Mid:* CIFAR10 (conditional); *Right:* ImageNet $64 \times 64$ (conditional)

Table 2: Comparison image generation on CIFAR-10 (unconditional). The best one/few-step generator under the FID metric is highlighted with **bold**. **F.S.** means *from scratch*. **L.T.** means *resume and Longer Training*.

| Family | Model | NFE | FID ($\downarrow$) |
|---|---|---|---|
| Teacher | VP-EDM [19] | 35 | 1.97 |
| Diffusion | DPM-Solver-3 [65] | 48 | 2.65 |
| | DDIM [46] | 100 | 4.16 |
| | DDPM [15] | 1000 | 3.17 |
| | NCSN++ [50] | 1000 | 2.38 |
| | iDDPM [38] | 4000 | 2.90 |
| Consistency | sCT [25] | 2 | 2.06 |
| | ECT [11] | 2 | 2.11 |
| | iCT [47] | 2 | 2.46 |
| Few Step | PD [43] | 2 | 4.51 |
| | IMM [66] | 2 | 1.98 |
| | TRACT [3] | 2 | 3.32 |
| | KD [26] | 1 | 9.36 |
| | Diff. ProjGAN [54] | 1 | 2.54 |
| | PID [52] | 1 | 3.92 |
| | DFNO [64] | 1 | 3.78 |
| | iCT-deep [47] | 1 | 2.51 |
| | Diff-Instruct [30] | 1 | 4.53 |
| | DMD [62] | 1 | 3.77 |
| | CTM [20] | 1 | 1.98 |
| | SiD [69] | 1 | 1.92 |
| | SiDA [68] | 1 | 1.52 |
| | SiD$^2$A [68] | 1 | 1.50 |
| | **Uni-Instruct with RKL (F.S.)** | 1 | 1.52 |
| | **Uni-Instruct with FKL (F.S.)** | 1 | 1.52 |
| | **Uni-Instruct with FKL (L.T.)** | 1 | 1.48 |
| | **Uni-Instruct with JKL (F.S.)** | 1 | **1.46** |

Table 3: Class conditional ImageNet $64 \times 64$ generation results. "Direct generation" and "Distillation" methods require one NFE, while the teacher uses 35 NFE. **F.S.** means *from scratch*. **L.T.** means *resume and Longer Training*.

| Family | Model | NFE | FID ($\downarrow$) |
|---|---|---|---|
| Teacher | VP-EDM [19] | 511 | 1.36 |
| Diffusion | RIN [18] | 1000 | 1.23 |
| | DDPM [15] | 250 | 11.00 |
| | ADM [9] | 250 | 2.07 |
| | DiT-L/2 [40] | 250 | 2.91 |
| | U-ViT [2] | 50 | 4.26 |
| Consistency | iCT [47] | 1 | 4.02 |
| | iCT-deep [47] | 1 | 3.25 |
| | ECT [11] | 1 | 2.49 |
| Few Step | MMD [44] | 8 | 1.24 |
| | G-istill [35] | 8 | 2.05 |
| | PD [43] | 2 | 8.95 |
| | Diff-Instruct [30] | 1 | 5.57 |
| | PID [52] | 1 | 9.49 |
| | iCT-deep [47] | 1 | 3.25 |
| | EMD-16 [57] | 1 | 2.20 |
| | DFNO [64] | 1 | 7.83 |
| | DMD2+longer training [61] | 1 | 1.28 |
| | CTM [20] | 1 | 1.92 |
| | SiD [69] | 1 | 1.71 |
| | SiDA [68] | 1 | 1.35 |
| | SiD$^2$A [68] | 1 | 1.10 |
| | $f$-distill [59] | 1 | 1.16 |
| | **Uni-Instruct with RKL(F.S.)** | 1 | 1.35 |
| | **Uni-Instruct with JKL(F.S.)** | 1 | 1.28 |
| | **Uni-Instruct with FKL(F.S.)** | 1 | 1.34 |
| | **Uni-Instruct with FKL(L.T.)** | 1 | **1.02** |

## 4.1 Benchmark Datasets Generation

**Experiment Settings**   We evaluate Uni-Instruct for both conditional and unconditional generations on CIFAR10 [22] and conditional generations on ImageNet $64 \times 64$[8]. We use EDM [19] as teacher models. In each experiment, we implement three types of divergences: Reverse-KL (RKL), Forward-KL (FKL), and Jeffrey-KL (JKL) divergence. We borrow the parameters settings from SiDA [68], which takes the output from the diffusion unet encoder directly as the discriminator. As for evaluation metrics, we use FID, as it simultaneously quantifies both image quality and diversity.

**Performance Evaluations**   Tab. 2, Tab. 4 and Tab. 3 shows Uni-Instruct performance on both settings of CIFAR10 and ImageNet $64 \times 64$. Uni-Instruct achieves new state-of-the-art one-step generation performances on all datasets. Our important findings include: (1) **When training from scratch, JKL achieves the lowest FID score.** On CIFAR10, JKL trained from scratch has a FID score of $1.42$, out-perform other baseline methods like DMD [62], SiDA [68], and the teacher model EDM [19]. (2) **When resuming a trained SiD model (RKL), FKL achieves even better results.** As is shown in the Table 2, FKL with longer training achieves a new state-of-the-art one-step generation on both datasets. This means a two-time training schedule: first trained with RKL until convergence, followed by FKL, enhances the model's performance with both mode-seeking behavior from RKL and mode-covering behavior from FKL.

Table 4: Label-conditioned image generation results on CIFAR-10. The best one/few-step generator under the FID metric is highlighted with **bold**.

| Family | Model | NFE | FID ($\downarrow$) |
|---|---|---|---|
| Teacher | VP-EDM [19] | 35 | 1.79 |
| Diffusion | DDPM [15] | 1000 | 3.17 |
| | iDDPM [38] | 4000 | 2.90 |
| One Step | Diff-Instruct [30] | 1 | 4.19 |
| | SIM [31] | 1 | 1.96 |
| | CTM [20] | 1 | 1.73 |
| | SiD [69] | 1 | 1.71 |
| | SiDA [68] | 1 | 1.44 |
| | SiD$^2$A [68] | 1 | 1.40 |
| | $f$-distill [59] | 1 | 1.92 |
| | **Uni-Instruct w. RKL (from scratch)** | 1 | 1.44 |
| | **Uni-Instruct w. JKL (from scratch)** | 1 | 1.42 |
| | **Uni-Instruct w. FKL (from scratch)** | 1 | 1.43 |
| | **Uni-Instruct w. FKL (longer training)** | 1 | **1.38** |

Table 5: Ablation study on CIFAR10 uncond generation. **GAN** means using GAN loss. **Init** means initialize from models.

| Div. | SiD Init. | GAN | FID$\downarrow$ |
|---|---|---|---|
| None | | $\checkmark$ | 8.21 |
| $\chi^2$ | | $\checkmark$ | 4.37 |
| JS | | $\checkmark$ | 5.23 |
| JKL | | $\checkmark$ | **1.46** |
| RKL | | | 1.92 |
| FKL | | | 1.88 |
| RKL | | $\checkmark$ | 1.52 |
| FKL | | $\checkmark$ | 1.52 |
| RKL | $\checkmark$ | $\checkmark$ | 1.50 |
| FKL | $\checkmark$ | $\checkmark$ | 1.48 |
| JKL | $\checkmark$ | $\checkmark$ | 1.50 |

## 4.2 Ablation Studies

**Performance Between Different Divergences and the effect of GAN loss.**   We perform an ablation study on the techniques applied in our experiments. Table 5 ablates different components of our proposed method on CIFAR10, where we use an unconditional generator for all settings. For different divergences, we select three types: JKL, FKL, and RKL are divergences that only contains Grad(SiD), $\chi^2$ divergence's gradient is only contributed by Grad(DI), Jensen-Shannon (JS) divergence has a gradient that contains both: $h_{\text{DI}}(\mathbf{x})\text{Grad(DI)} + h_{\text{SiD}}(\mathbf{x})\text{Grad(SiD)}$. Our result shows that JKL achieves the lowest FID value. Due to the numerical instability of the weightings, JS yields unsuccessful distillation results. As for the effect of GAN loss, we find that removing it still yields a decent result. Our integrated approach also surpasses the performance of using Uni-Instruct loss alone(without adding GAN loss), highlighting the effectiveness of combining expanded $f$-divergence with GAN losses. We also find that using a model trained with RKL Uni-Instruct (which recovers the SiD[69] loss) as the initialization leads to better performances for all divergences.

**Additional metrics evaluation and convergence analysis.**   As is shown in the right table of Tab. 6, across different training iterations, our method consistently achieves lower FID scores than SiDA. This clearly indicates that our approach converges faster, reaching better generative quality with fewer iterations. Moreover, Tab. 6 and Tab. 7 compares the performance of Uni-Instruct and SiDA [68] across both CIFAR10 and ImageNet $64 \times 64$ benchmarks on sFID, FD$_{\text{DINO}}$, inception score (IS), percision and recall. Uni-Instruct achieves strictly better results in 4 of 5 metrics on ImageNet $64 \times 64$, the more complex and practically relevant benchmark. These gains are significant, particularly in FD$_{\text{dino}}$ and IS, which measure semantic alignment and perceptual quality/diversity, respectively.

The 18.7% reduction in $FD_{dino}$ on ImageNet $64 \times 64$ confirms Uni-Instruct's advanced capability to preserve high-level semantic structures (e.g., object boundaries, textures, contextual relationships). This is critical for applications requiring fine-grained realism (e.g., medical imaging, autonomous driving). More importantly, Uni-Instruct's gains widen significantly on ImageNet $64 \times 64$ ($+8.1\%$ IS, $-18.7\%$ $FD_{dino}$) versus CIFAR10, proving its robustness for high resolution, semantically rich image synthesis. SiDA fails to maintain competitiveness under greater complexity.

Table 6: Further comparison between SiDA[68] and Uni-Instruct (forward KL). The left table is the performance evaluation on CIFAR10 unconditional generation. The right table compares the FID score along with the iterated k-images during training.

| Method | sFID↓ | $FD_{dino}$ ↓ | IS↑ | Precision↑ | Recall↑ | $10^1$ | $10^2$ | $10^3$ | $10^4$ | $10^5$ |
|--------|-------|---------------|-----|------------|---------|--------|--------|--------|--------|--------|
| SiDA | 1.71 | 132.72 | 10.32 | 0.670 | 0.624 | 139.86 | 68.73 | 41.45 | 6.98 | 1.44 |
| Ours | 1.66 | 129.30 | 10.30 | 0.671 | 0.626 | 132.51 | 54.60 | 38.58 | 5.23 | 1.41 |

| | sFID↓ | $FD_{dino}$ ↓ | IS↑ | Precision↑ | Recall↑ | sFID↓ | $FD_{dino}$ ↓ | IS↑ | Precision↑ | Recall↑ |
|------|-------|---------------|-----|------------|---------|-------|---------------|-----|------------|---------|
| SiDA | 1.68 | 111.26 | 10.28 | 0.678 | 0.632 | 1.98 | 74.86 | 59.28 | 0.562 | 0.653 |
| Ours | 1.68 | 108.89 | 10.29 | 0.679 | 0.629 | 2.01 | 60.86 | 64.11 | 0.561 | 0.658 |

Table 7: Further comparison between SiDA[68] and Uni-Instruct (forward KL). The left table is the performance evaluation on CIFAR10 conditional generation, while the right table is the performance evaluation on ImageNet $64\times64$ generation.

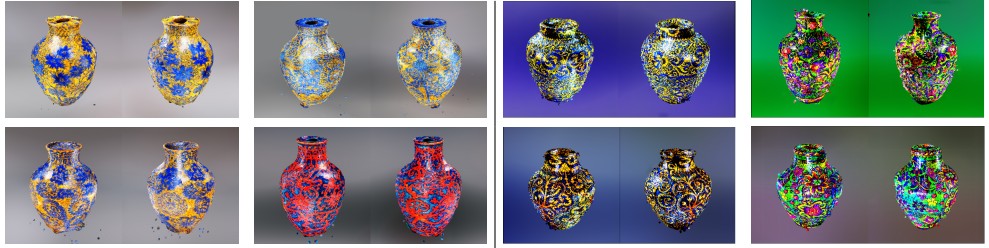

Figure 3: **Prompt**: A refined vase with artistic patterns. Left: ProlificDreamer; Right: Uni-Instruct (forward KL). Our vase demonstrates more diverse shapes as well as realistic patterns.

### 4.3 Text-to-3D Generation Using 2D Diffusion

In this subsection, we apply Uni-Instruct on text-to-3D generation. We re-implement the code base of ProlificDreamer [55] by adding an extra discriminator head to the output of the stable diffusion Unet's encoder. We use FKL to distill the model for 400 epochs. Fig. 3 demonstrates the visual results from our 3D experiments. Uni-Instruct archives surprisingly decent 3D generation performances, with improved diversity and fidelity. Due to page limitations, we put detailed experiment settings, quantitative metrics, and training algorithm in the Appendix E.

## 5 Conclusions

We present Uni-Instruct, a theoretically grounded framework for training one-step diffusion models via distribution matching. Through building upon a novel *diffusion expansion* theory of the $f$-divergence, Uni-Instruct establishes a unifying theoretical foundation that generalizes and connects more than 10 existing diffusion distillation methodologies. Uni-Instruct also demonstrates superior performance on benchmark datasets and efficacy in downstream tasks like text-to-3D generation. We hope Uni-Instruct offers useful insights for future studies on efficient generative models.

## Acknowledgement

This work was supported by the National Natural Science Foundation of China (62371007) and by Xiaohongshu Inc. The authors acknowledge helpful advice from Shanghai AI Lab and Yongqian Peng.

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

## A    Related Works

**Diffusion Distillation**    Diffusion distillation [28] focuses on reducing generation costs by transferring knowledge from teacher diffusion models to more efficient student models. It primarily includes three categories of methods: (1) *Trajectory Distillation:* These methods train student models to approximate the generation trajectory of diffusion models using fewer denoising steps. Approaches such as direct distillation [26, 10] and progressive distillation [43, 35] aim to predict cleaner data from noisy inputs. Consistency-based methods [48, 20, 47, 23, 12] instead minimize a self-consistency loss across intermediate steps. Most of these methods require access to real data samples for effective training. (2) *Divergence Minimization (Distribution Matching):* This line of work aims to align the distribution of the student model with that of the teacher. Adversarial training-based methods [56, 58] typically require real data to perform distribution matching. Alternatively, several approaches minimize divergences like the KL divergence (e.g., Diff-Instruct [30, 62]) or Fisher divergence (e.g., Score Identity Distillation [69], Score Implicit Matching [31]), and often do so without requiring real samples. Numerous improvements have been made to these two lines of work: DMD2 [61] and SiDA [68] add real images during training, rapidly surpassing the teacher's performance. $f$-distill [59] generalize KL divergence of Diff-Instruct into $f$-divergence and compared the affection of different divergences. Additionally, significant progress has been made toward scaling diffusion distillation for ultra-fast or even one-step text-to-image generation [27, 16, 51, 62, 67, 61]. (3) *Other Methods:* Several alternative techniques that train the model from scratch have been proposed, including ReFlow [24], Flow Matching Models (FMM) [4], which propose an ODE to model the diffusion process. Inductive Moment Matching [66] models the self-consistency of stochastic interpolants at different time steps. Consistency models [48, 47, 20, 11] impose consistency constraints on network outputs along the trajectory.

## B    Proofs

### B.1    Proof of Theorem 3.1

*Proof.* Let $p_t$ and $q_t$ be distributions satisfying the Fokker-Planck equations, and decay rapidly at infinity:

$$\frac{\partial p_{\theta,t}}{\partial t} = \nabla_{\boldsymbol{x}} \cdot \left[ \frac{1}{2} g^2(t) p_{\theta,t} \nabla_{\boldsymbol{x}} \log p_{\theta,t} - \boldsymbol{F}(\boldsymbol{x}, t) p_{\theta,t} \right]$$

$$\frac{\partial q_t}{\partial t} = \nabla_{\boldsymbol{x}} \cdot \left[ \frac{1}{2} g^2(t) q_t \nabla_{\boldsymbol{x}} \log q_t - \boldsymbol{F}(\boldsymbol{x}, t) q_t \right] \tag{B.1}$$

We begin with the definition of $f$-divergence and apply differentiation under the integral sign:

$$\frac{\mathrm{d}}{\mathrm{d}t} \mathcal{D}_f(q_t \| p_{\theta,t}) = \frac{\mathrm{d}}{\mathrm{d}t} \int p_{\theta,t} f\left( \frac{q_t}{p_{\theta,t}} \right) \mathrm{d}\boldsymbol{x}$$

$$= \int \frac{\partial p_{\theta,t}}{\partial t} f\left( \frac{q_t}{p_{\theta,t}} \right) \mathrm{d}\boldsymbol{x} + \int p_{\theta,t} \frac{\partial}{\partial t} f\left( \frac{q_t}{p_{\theta,t}} \right) \mathrm{d}\boldsymbol{x} \tag{B.2}$$

For the second term, apply the chain rule and the quotient rule:

$$\int p_{\theta,t} \frac{\partial}{\partial t} f\left( \frac{q_t}{p_{\theta,t}} \right) \mathrm{d}\boldsymbol{x} = \int p_{\theta,t} f'\left( \frac{q_t}{p_{\theta,t}} \right) \frac{\partial}{\partial t} \left( \frac{q_t}{p_{\theta,t}} \right) \mathrm{d}\boldsymbol{x}$$

$$= \int f'\left( \frac{q_t}{p_{\theta,t}} \right) \left( \frac{\partial q_t}{\partial t} - \frac{q_t}{p_{\theta,t}} \frac{\partial p_{\theta,t}}{\partial t} \right) \mathrm{d}\boldsymbol{x} \tag{B.3}$$

Combining Eq. B.1, Eq. B.2 and Eq. B.3, we obtain:

$$\frac{\mathrm{d}}{\mathrm{d}t} \mathcal{D}_f(q_t \| p_{\theta,t}) = \int \nabla_{\boldsymbol{x}} \left[ \frac{1}{2} g^2(t) p_{\theta,t} \nabla_{\boldsymbol{x}} \log p_{\theta,t} - \boldsymbol{F}(\boldsymbol{x}, t) p_{\theta,t} \right] f\left( \frac{q_t}{p_{\theta,t}} \right) \mathrm{d}\boldsymbol{x}$$

$$+ \int f'\left( \frac{q_t}{p_{\theta,t}} \right) \nabla_{\boldsymbol{x}} \left[ \frac{1}{2} g^2(t) q_t \nabla_{\boldsymbol{x}} \log q_t - \boldsymbol{F}(\boldsymbol{x}, t) q_t \right] \mathrm{d}\boldsymbol{x}$$

$$- \int \frac{q_t}{p_{\theta,t}} f'\left( \frac{q_t}{p_{\theta,t}} \right) \nabla_{\boldsymbol{x}} \left[ \frac{1}{2} g^2(t) p_{\theta,t} \nabla_{\boldsymbol{x}} \log p_{\theta,t} - \boldsymbol{F}(\boldsymbol{x}, t) p_{\theta,t} \right] \mathrm{d}\boldsymbol{x} \tag{B.4}$$

Apply integration by parts to the RHS of Eq. B.4 and with previous assumption that distribution $p_{\theta,t}$ and $q_t$ decay rapidly at infinity, we have:

$$\frac{\mathrm{d}}{\mathrm{d}t}\mathcal{D}_f(q_t\|p_{\theta,t}) = -\int \left[\frac{1}{2}g^2(t)p_{\theta,t}\nabla_x\log p_{\theta,t} - \boldsymbol{F}(\boldsymbol{x},t)p_{\theta,t}\right]\nabla_x f\left(\frac{q_t}{p_{\theta,t}}\right)\mathrm{d}\boldsymbol{x}$$
$$-\int \nabla_x f'\left(\frac{q_t}{p_{\theta,t}}\right)\left[\frac{1}{2}g^2(t)q_t\nabla_x\log q_t - \boldsymbol{F}(\boldsymbol{x},t)q_t\right]\mathrm{d}\boldsymbol{x}$$
$$+\int \nabla_x\left[\frac{q_t}{p_{\theta,t}}f'\left(\frac{q_t}{p_{\theta,t}}\right)\right]\left[\frac{1}{2}g^2(t)p_{\theta,t}\nabla_x\log p_{\theta,t} - \boldsymbol{F}(\boldsymbol{x},t)p_{\theta,t}\right]\mathrm{d}\boldsymbol{x} \quad \text{(B.5)}$$

Now we can further expand the gradient terms in Eq. B.5:

$$\nabla_{\boldsymbol{x}}f\left(\frac{q_t}{p_{\theta,t}}\right) = f'\left(\frac{q_t}{p_{\theta,t}}\right)\frac{\nabla_{\boldsymbol{x}}q_t p_{\theta,t} - \nabla_{\boldsymbol{x}}p_{\theta,t}q_t}{p_{\theta,t}^2} \quad \text{(B.6)}$$

$$\nabla_{\boldsymbol{x}}f'\left(\frac{q_t}{p_{\theta,t}}\right) = f''\left(\frac{q_t}{p_{\theta,t}}\right)\frac{\nabla_{\boldsymbol{x}}q_t p_{\theta,t} - \nabla_{\boldsymbol{x}}p_{\theta,t}q_t}{p_{\theta,t}^2} \quad \text{(B.7)}$$

$$\nabla_{\boldsymbol{x}}\left[\frac{q_t}{p_{\theta,t}}f'\left(\frac{q_t}{p_{\theta,t}}\right)\right] = f'\left(\frac{q_t}{p_{\theta,t}}\right)\frac{\nabla_{\boldsymbol{x}}q_t p_{\theta,t} - \nabla_{\boldsymbol{x}}p_{\theta,t}q_t}{p_{\theta,t}^2} + \frac{q_t}{p_{\theta,t}}f''\left(\frac{q_t}{p_{\theta,t}}\right)\frac{\nabla_{\boldsymbol{x}}q_t p_{\theta,t} - \nabla_{\boldsymbol{x}}p_{\theta,t}q_t}{p_{\theta,t}^2}$$
$$\text{(B.8)}$$

Replace the gradient terms in Eq. B.5 with Eq. B.6, Eq. B.7, and Eq. B.8 and after algebraic manipulation, we obtain:

$$\frac{\mathrm{d}}{\mathrm{d}t}\mathcal{D}_f(q_t\|p_{\theta,t}) = -\frac{1}{2}g^2(t)\int p_{\theta,t}\left(\frac{q_t}{p_{\theta,t}}\right)^2 f''\left(\frac{q_t}{p_{\theta,t}}\right)\|\nabla_{\boldsymbol{x}}\log q_t - \nabla_{\boldsymbol{x}}\log p_{\theta,t}\|^2\mathrm{d}\boldsymbol{x} \quad \text{(B.9)}$$

The integral version of Eq. B.9 is:

$$\mathcal{D}_f(q_0\|p_\theta) = \int_0^T \frac{1}{2}g^2(t)\mathbb{E}_{p_{\theta,t}}\left[\left(\frac{q_t}{p_{\theta,t}}\right)^2 f''\left(\frac{q_t}{p_{\theta,t}}\right)\|\nabla_{\boldsymbol{x}}\log q_t - \nabla_{\boldsymbol{x}}\log p_{\theta,t}\|^2\right]\mathrm{d}t + \mathcal{D}_f(q_T\|p_{\theta,T})$$
$$\text{(B.10)}$$

$\square$

## B.2   Proof of Theorem 3.2

**Lemma B.1** (Calculate the gradient of $\boldsymbol{x} \sim p_{\theta,t}$ [59])*. Assuming that sampling from $\boldsymbol{x} \sim p_{\theta,t}$ can be parameterized as $\boldsymbol{x} = G_\theta(\boldsymbol{z}) + \sigma(t)\epsilon$, where $\boldsymbol{z} \sim p(\boldsymbol{z})$, $\epsilon \sim \mathcal{N}(0, I)$, and $G_\theta$, $g$ are differentiable mappings. In addition, $g$ is constant with respect to $\theta$. Then,*

$$\int \nabla_\theta p_{\theta,t}(\boldsymbol{x})g(\boldsymbol{x})\,\mathrm{d}\boldsymbol{x} = \int\int p(\epsilon)p(\boldsymbol{z})\nabla_{\boldsymbol{x}}g(\boldsymbol{x})\nabla_\theta G_\theta(\boldsymbol{z})\,\mathrm{d}\epsilon\,\mathrm{d}\boldsymbol{z}.$$

*Proof.* As $q_t$ and $g$ are both continuous functions, we can interchange integration and differentiation:

$$\int \nabla_\theta p_{\theta,t}(\boldsymbol{x})g(\boldsymbol{x})\,\mathrm{d}\boldsymbol{x} = \nabla_\theta \int p_{\theta,t}(\boldsymbol{x})g(\boldsymbol{x})\,\mathrm{d}\boldsymbol{x}$$
$$= \int\int p(\epsilon)p(\boldsymbol{z})\nabla_\theta g(G_\theta(\boldsymbol{z}) + \sigma(t)\epsilon)\,\mathrm{d}\epsilon\,\mathrm{d}\boldsymbol{z}$$
$$= \int\int p(\epsilon)p(\boldsymbol{z})\nabla_{\boldsymbol{x}}g(\boldsymbol{x})\nabla_\theta G_\theta(\boldsymbol{z})\,\mathrm{d}\epsilon\,\mathrm{d}\boldsymbol{z}$$
$$= \int p_{\theta,t}(\boldsymbol{x})\nabla_{\boldsymbol{x}}g(\boldsymbol{x})\frac{\partial\boldsymbol{x}}{\partial\theta}\,\mathrm{d}\boldsymbol{x},$$

where $\boldsymbol{x} = G_\theta(\boldsymbol{z}) + \sigma(t)\epsilon$. $\square$

**Lemma B.2** (Calculate the gradient of the score fuction [31])**.** *If distribution $p_{\theta,t}$ satisfies some mild regularity conditions, we have for any score function $s_{q_t}(\cdot)$, the following equation holds for all parameter $\theta$:*

$$\mathbb{E}_{\boldsymbol{x}_t \sim p_{sg[\theta],t}} \left[ \left( s_{p_\theta,t}(\boldsymbol{x}_t) - s_{q_t}(\boldsymbol{x}_t) \right) \frac{\partial}{\partial \theta} s_{p_\theta,t}(\boldsymbol{x}_t) \right] \tag{B.11}$$

$$= -\frac{\partial}{\partial \theta} \mathbb{E} \left[ \left\{ \left( s_{sg[\theta],t}(\boldsymbol{x}_t) - s_{q_t}(\boldsymbol{x}_t) \right) \right\}^T \left\{ s_{sg[\theta],t}(\boldsymbol{x}_t) - \nabla_{\boldsymbol{x}_t} \log q_t(\boldsymbol{x}_t|\boldsymbol{x}_0) \right\} \right] \tag{B.12}$$

For completeness, we appreciate the efforts of Luo et al. [31] and provide the proof here. The original version can be refered to Theorem 3.1 from [31].

*Proof.* Starting with score projection identity [69]:

$$\mathbb{E}_{\substack{\boldsymbol{x}_0 \sim p_\theta,0 \\ \boldsymbol{x}_t|\boldsymbol{x}_0 \sim q_t(\boldsymbol{x}_t|\boldsymbol{x}_0)}} \left\{ u(\boldsymbol{x}_t,\theta)^T \left( s_{p_\theta,t}(\boldsymbol{x}_t) - \nabla_{\boldsymbol{x}_t} \log q_t(\boldsymbol{x}_t|\boldsymbol{x}_0) \right) \right\} = 0, \quad \forall \theta, \forall u. \tag{B.13}$$

Taking the gradient with respect to $\theta$ on the above identity, we have:

$$0 = \mathbb{E}_{\substack{\boldsymbol{x}_0 \sim p_\theta,0 \\ \boldsymbol{x}_t|\boldsymbol{x}_0 \sim q_t(\boldsymbol{x}_t|\boldsymbol{x}_0)}} \left\{ \frac{\partial}{\partial \boldsymbol{x}_t} \left( u(\boldsymbol{x}_t,\theta)^T \left\{ s_{p_\theta,t}(\boldsymbol{x}_t) - \nabla_{\boldsymbol{x}_t} \log q_t(\boldsymbol{x}_t|\boldsymbol{x}_0) \right\} \right) \right\} \frac{\partial \boldsymbol{x}_t}{\partial \theta} \tag{B.14}$$

$$+ \mathbb{E}_{\substack{\boldsymbol{x}_0 \sim p_\theta,0 \\ \boldsymbol{x}_t|\boldsymbol{x}_0 \sim q_t(\boldsymbol{x}_t|\boldsymbol{x}_0)}} \left\{ \frac{\partial}{\partial \boldsymbol{x}_0} \left( u(\boldsymbol{x}_t,\theta)^T \left\{ -\nabla_{\boldsymbol{x}_t} \log q_t(\boldsymbol{x}_t|\boldsymbol{x}_0) \right\} \right) \right\} \frac{\partial \boldsymbol{x}_0}{\partial \theta} \tag{B.15}$$

$$+ \mathbb{E}_{\substack{\boldsymbol{x}_0 \sim p_\theta,0 \\ \boldsymbol{x}_t|\boldsymbol{x}_0 \sim q_t(\boldsymbol{x}_t|\boldsymbol{x}_0)}} \left\{ u(\boldsymbol{x}_t,\theta)^T \frac{\partial}{\partial \theta} \left\{ s_{p_\theta,t}(\boldsymbol{x}_t) \right\} \right\} + \frac{\partial}{\partial \theta} u(\boldsymbol{x}_t,\theta)^T s_\theta(\boldsymbol{x}_t) \tag{B.16}$$

$$= \mathbb{E}_{\substack{\boldsymbol{x}_0 \sim p_\theta,0 \\ \boldsymbol{x}_t|\boldsymbol{x}_0 \sim q_t(\boldsymbol{x}_t|\boldsymbol{x}_0)}} \left\{ u(\boldsymbol{x}_t,\theta)^T \frac{\partial}{\partial \theta} \left\{ s_{p_\theta,t}(\boldsymbol{x}_t) \right\} \right\} \tag{B.17}$$

$$+ \mathbb{E}_{\substack{\boldsymbol{x}_0 \sim p_\theta,0 \\ \boldsymbol{x}_t|\boldsymbol{x}_0 \sim q_t(\boldsymbol{x}_t|\boldsymbol{x}_0)}} \left\{ \frac{\partial}{\partial \boldsymbol{x}_t} \left\{ \frac{\partial}{\partial \theta} \left( u(\boldsymbol{x}_t,\theta)^T \left\{ s_{p_\theta,t}(\boldsymbol{x}_t) - \nabla_{\boldsymbol{x}_t} \log q_t(\boldsymbol{x}_t|\boldsymbol{x}_0) \right\} \right) \right\} \frac{\partial \boldsymbol{x}_t}{\partial \theta} \right\} \tag{B.18}$$

$$+ \mathbb{E}_{\substack{\boldsymbol{x}_0 \sim p_\theta,0 \\ \boldsymbol{x}_t \boldsymbol{x}_0 \sim q_t(\boldsymbol{x}_t|\boldsymbol{x}_0)}} \left\{ \frac{\partial}{\partial \boldsymbol{x}_0} \left\{ u(\boldsymbol{x}_t,\theta)^T \left\{ -\nabla_{\boldsymbol{x}_t} \log q_t(\boldsymbol{x}_t|\boldsymbol{x}_0) \right\} \right\} \frac{\partial \boldsymbol{x}_0}{\partial \theta} + \frac{\partial}{\partial \theta} u(\boldsymbol{x}_t,\theta)^T s_\theta(\boldsymbol{x}_t) \right\} \tag{B.19}$$

$$= \mathbb{E}_{\boldsymbol{x}_t \sim p_\theta,t} \left\{ u(\boldsymbol{x}_t,\theta)^T \frac{\partial}{\partial \theta} \left\{ s_{p_\theta,t}(\boldsymbol{x}_t) \right\} \right\} \tag{B.20}$$

$$+ \frac{\partial}{\partial \theta} \mathbb{E}_{\substack{\boldsymbol{x}_0 \sim p_\theta,0 \\ \boldsymbol{x}_t|\boldsymbol{x}_0 \sim q_t(\boldsymbol{x}_t|\boldsymbol{x}_0)}} \left\{ u(\boldsymbol{x}_t,\theta)^T \left\{ s_{p_\theta,t}(\boldsymbol{x}_t) - \nabla_{\boldsymbol{x}_t} \log q_t(\boldsymbol{x}_t|\boldsymbol{x}_0) \right\} \right\}. \tag{B.21}$$

Therefore, we obtain the following identity:

$$\mathbb{E}_{\boldsymbol{x}_t \sim p_\theta,t} \left\{ u(\boldsymbol{x}_t,\theta)^T \frac{\partial}{\partial \theta} s_{p_\theta,t}(\boldsymbol{x}_t) \right\} = -\frac{\partial}{\partial \theta} \mathbb{E}_{\substack{\boldsymbol{x}_0 \sim p_\theta,0 \\ \boldsymbol{x}_t|\boldsymbol{x}_0 \sim q_t(\boldsymbol{x}_t|\boldsymbol{x}_0)}} \left\{ u(\boldsymbol{x}_t,\theta)^T \left( s_{p_\theta,t}(\boldsymbol{x}_t) - \nabla_{\boldsymbol{x}_t} \log q_t(\boldsymbol{x}_t|\boldsymbol{x}_0) \right) \right\}. \tag{B.22}$$

Replacing $u(\boldsymbol{x}_t)$ with $s_{p_\theta,t}(\boldsymbol{x}_t) - s_{q_t}(\boldsymbol{x}_t)$ we can proof the correctness of the original identity.

$\square$

We now complete the proof of Theorem 3.2:

*Proof.* Applying the product rule to the gradient, we can obtain:

$$\nabla_\theta \left\{ \frac{1}{2} g^2(t) \mathbb{E}_{p_{\theta,t}} \left[ \left( \frac{q_t}{p_{\theta,t}} \right)^2 f'' \left( \frac{q_t}{p_{\theta,t}} \right) \| \nabla \log p_{\theta,t} - \nabla \log q_t \|_2^2 \right] \right\} \tag{B.23}$$

$$= \frac{1}{2} g^2(t) \nabla_\theta \int p_{\theta,t}(\boldsymbol{x}_t) \left( \frac{q_t}{p_{\theta,t}} \right)^2 f'' \left( \frac{q_t}{p_{\theta,t}} \right) \| \nabla \log p_{\theta,t} - \nabla \log q_t \|_2^2 \mathrm{d}\boldsymbol{x}_t \tag{B.24}$$

$$= \frac{1}{2} g^2(t) \int \nabla_\theta p_{\theta,t}(\boldsymbol{x}_t) \left( \frac{q_t}{p_{\theta,t}} \right)^2 f'' \left( \frac{q_t}{p_{\theta,t}} \right) \| \nabla \log p_{\theta,t} - \nabla \log q_t \|_2^2 \mathrm{d}\boldsymbol{x}_t \tag{B.25}$$

$$+ \frac{1}{2} g^2(t) \int p_{\theta,t}(\boldsymbol{x}_t) \nabla_\theta \left[ \left( \frac{q_t}{p_{\theta,t}} \right)^2 f'' \left( \frac{q_t}{p_{\theta,t}} \right) \| \nabla \log p_{\theta,t} - \nabla \log q_t \|_2^2 \right] \mathrm{d}\boldsymbol{x}_t, \tag{B.26}$$

which can be further decomposed into the following four terms:

$$\mathrm{Grad} = \underbrace{\frac{1}{2} g^2(t) \int \nabla_\theta p_{\theta,t}(\boldsymbol{x}_t) \left( \frac{q_t}{p_{\theta,t}} \right)^2 f'' \left( \frac{q_t}{p_{\theta,t}} \right) \| \nabla \log p_{\theta,t} - \nabla \log q_t \|_2^2 \mathrm{d}\boldsymbol{x}_t}_{A} \tag{B.27}$$

$$+ \underbrace{\frac{1}{2} g^2(t) \int p_{\theta,t}(\boldsymbol{x}_t) \nabla_\theta \left[ \left( \frac{q_t}{p_{\theta,t}} \right)^2 \right] f'' \left( \frac{q_t}{p_{\theta,t}} \right) \| \nabla \log p_{\theta,t} - \nabla \log q_t \|_2^2 \mathrm{d}\boldsymbol{x}_t}_{B} \tag{B.28}$$

$$+ \underbrace{\frac{1}{2} g^2(t) \int p_{\theta,t}(\boldsymbol{x}_t) \left( \frac{q_t}{p_{\theta,t}} \right)^2 \nabla_\theta \left[ f'' \left( \frac{q_t}{p_{\theta,t}} \right) \right] \| \nabla \log p_{\theta,t} - \nabla \log q_t \|_2^2 \mathrm{d}\boldsymbol{x}_t}_{C} \tag{B.29}$$

$$+ \underbrace{\frac{1}{2} g^2(t) \int p_{\theta,t}(\boldsymbol{x}_t) \left( \frac{q_t}{p_{\theta,t}} \right)^2 f'' \left( \frac{q_t}{p_{\theta,t}} \right) \nabla_\theta \left[ \| \nabla \log p_{\theta,t} - \nabla \log q_t \|_2^2 \right] \mathrm{d}\boldsymbol{x}_t}_{D} \tag{B.30}$$

We calculate the above four terms separately.

$$A = \frac{1}{2} g^2(t) \int \nabla_\theta p_{\theta,t}(\boldsymbol{x}_t) \left( \frac{q_t}{p_{\theta,t}} \right)^2 f'' \left( \frac{q_t}{p_{\theta,t}} \right) \| \nabla \log p_{\theta,t} - \nabla \log q_t \|_2^2 \mathrm{d}\boldsymbol{x}_t \tag{B.31}$$

$$= \frac{1}{2} g^2(t) \int p_{\theta,t}(\boldsymbol{x}_t) \left( 2 \frac{q_t}{p_{\theta,t}} \nabla_{\boldsymbol{x}} \frac{q_t}{p_{\theta,t}} \frac{\partial \boldsymbol{x}_t}{\partial \theta} \right) f'' \left( \frac{q_t}{p_{\theta,t}} \right) \| \nabla \log p_{\theta,t} - \nabla \log q_t \|_2^2 \mathrm{d}\boldsymbol{x}_t \tag{B.32}$$

$$+ \frac{1}{2} g^2(t) \int p_{\theta,t}(\boldsymbol{x}_t) \left( \frac{q_t}{p_{\theta,t}} \right)^2 \left( f''' \left( \frac{q_t}{p_{\theta,t}} \right) \nabla_{\boldsymbol{x}} \frac{q_t}{p_{\theta,t}} \frac{\partial \boldsymbol{x}_t}{\partial \theta} \right) \| \nabla \log p_{\theta,t} - \nabla \log q_t \|_2^2 \mathrm{d}\boldsymbol{x}_t \tag{B.33}$$

$$+ \frac{1}{2} g^2(t) \int p_{\theta,t}(\boldsymbol{x}_t) \left( \frac{q_t}{p_{\theta,t}} \right)^2 f'' \left( \frac{q_t}{p_{\theta,t}} \right) \nabla_\theta \left( \| \nabla \log p_{\theta,t} - \nabla \log q_t \|_2^2 \right) \mathrm{d}\boldsymbol{x}_t \tag{B.34}$$

$$B = \frac{1}{2} g^2(t) \int p_{\theta,t}(\boldsymbol{x}_t) \nabla_\theta \left[ \left( \frac{q_t}{p_{\theta,t}} \right)^2 \right] f'' \left( \frac{q_t}{p_{\theta,t}} \right) \| \nabla \log p_{\theta,t} - \nabla \log q_t \|_2^2 \mathrm{d}\boldsymbol{x}_t \tag{B.35}$$

$$= \frac{1}{2} g^2(t) \int p_{\theta,t}(\boldsymbol{x}_t) \left[ 2 \left( \frac{q_t}{p_{\theta,t}} \right) \left( -\frac{q_t}{p_{\theta,t}^2} \right) \nabla_\theta p_{\theta,t}(\boldsymbol{x}_t) \right] f'' \left( \frac{q_t}{p_{\theta,t}} \right) \| \nabla \log p_{\theta,t} - \nabla \log q_t \|_2^2 \mathrm{d}\boldsymbol{x}_t \tag{B.36}$$

$$= -\frac{1}{2} g^2(t) \int \nabla_\theta p_{\theta,t}(\boldsymbol{x}_t) \left[ 2 \left( \frac{q_t}{p_{\theta,t}} \right)^2 \right] f'' \left( \frac{q_t}{p_{\theta,t}} \right) \| \nabla \log p_{\theta,t} - \nabla \log q_t \|_2^2 \mathrm{d}\boldsymbol{x}_t \tag{B.37}$$

$$= -2 * A \tag{B.38}$$

$$C = \frac{1}{2}g^2(t) \int p_{\theta,t}(\boldsymbol{x}_t) \left(\frac{q_t}{p_{\theta,t}}\right)^2 \nabla_\theta \left[f''\left(\frac{q_t}{p_{\theta,t}}\right)\right] \|\nabla \log p_{\theta,t} - \nabla \log q_t\|_2^2 \mathrm{d}\boldsymbol{x}_t \tag{B.39}$$

$$= -\frac{1}{2}g^2(t) \int p_{\theta,t}(\boldsymbol{x}_t) \left(3\left(\frac{q_t}{p_{\theta,t}}\right)^2 \nabla_x \frac{q_t}{p_{\theta,t}} \frac{\partial \boldsymbol{x}_t}{\partial \theta}\right) f'''\left(\frac{q_t}{p_{\theta,t}}\right) \|\nabla \log p_{\theta,t} - \nabla \log q_t\|_2^2 \mathrm{d}\boldsymbol{x}_t \tag{B.40}$$

$$-\frac{1}{2}g^2(t) \int p_{\theta,t}(\boldsymbol{x}_t) \left(\frac{q_t}{p_{\theta,t}}\right)^3 \left(f''''\left(\frac{q_t}{p_{\theta,t}}\right) \nabla_x \frac{q_t}{p_{\theta,t}} \frac{\partial \boldsymbol{x}_t}{\partial \theta}\right) \|\nabla \log p_{\theta,t} - \nabla \log q_t\|_2^2 \mathrm{d}\boldsymbol{x}_t \tag{B.41}$$

$$-\frac{1}{2}g^2(t) \int p_{\theta,t}(\boldsymbol{x}_t) \left(\frac{q_t}{p_{\theta,t}}\right)^3 f'''\left(\frac{q_t}{p_{\theta,t}}\right) \nabla_\theta \left(\|\nabla \log p_{\theta,t} - \nabla \log q_t\|_2^2\right) \mathrm{d}\boldsymbol{x}_t \tag{B.42}$$

$$D = \frac{1}{2}g^2(t) \int p_{\theta,t}(\boldsymbol{x}_t) \left(\frac{q_t}{p_{\theta,t}}\right)^2 f''\left(\frac{q_t}{p_{\theta,t}}\right) \nabla_\theta \left[\|\nabla \log p_{\theta,t} - \nabla \log q_t\|_2^2\right] \mathrm{d}\boldsymbol{x}_t \tag{B.43}$$

As a result:

$$\nabla_\theta \left\{\frac{1}{2}g^2(t)\mathbb{E}_{p_{\theta,t}} \left[\left(\frac{q_t}{p_{\theta,t}}\right)^2 f''\left(\frac{q_t}{p_{\theta,t}}\right) \|\nabla \log p_{\theta,t} - \nabla \log q_t\|_2^2\right]\right\} \tag{B.44}$$

$$= A + B + C + D = -A + C + D \tag{B.45}$$

$$= \frac{1}{2}g^2(t)\mathbb{E}_{p_{\theta,t}} \left[\underbrace{\left[\left(\frac{q_t}{p_{\theta,t}}\right)^3 f'''\left(\frac{q_t}{p_{\theta,t}}\right)\right]}_{\text{weight 1}} \nabla_\theta \|\nabla \log p_{\theta,t} - \nabla \log q_t\|_2^2\right] \tag{B.46}$$

$$+ \frac{1}{2}g^2(t)\mathbb{E}_{p_{\theta,t}} \left[\underbrace{(*) \|\nabla \log p_{\theta,t} - \nabla \log q_t\|_2^2}_{\text{weight 2}} (\nabla \log p_{\theta,t} - \nabla \log q_t) \frac{\partial \boldsymbol{x}_t}{\partial \theta}\right] \tag{B.47}$$

where $(*)$ stands for $2\left(\frac{q_t}{p_{\theta,t}}\right)^2 f''\left(\frac{q_t}{p_{\theta,t}}\right) + 4\left(\frac{q_t}{p_{\theta,t}}\right)^3 f'''\left(\frac{q_t}{p_{\theta,t}}\right) + \left(\frac{q_t}{p_{\theta,t}}\right)^4 f''''\left(\frac{q_t}{p_{\theta,t}}\right)$.

Now we will focus on the only intractable term left in the previous equation: $\nabla_\theta \|\nabla \log p_{\theta,t} - \nabla \log q_t\|_2^2$. Such a problem has been well studied by FGM [17] and SIM[31]. The former one calculated the term under the assumption that $\boldsymbol{x}$ has gradient dependence on $\theta$, while SIM [31] simply ignores such an assumption and achieves comparable performance. For the simplicity of the loss expression, we follow the setting in SIM[31]. Thus we calculate $2(\nabla \log p_{\theta,t} - \nabla \log q_t)\frac{\partial \boldsymbol{s}_{p_{\theta,t}}(\boldsymbol{x}_t)}{\partial \theta}$.

Applying Lemma B.2, we have:

$$\frac{1}{2}g^2(t)\nabla_\theta \left\{\mathbb{E}_{p_{\theta,t}} \left[\left(\frac{q_t}{p_{\theta,t}}\right)^2 f''\left(\frac{q_t}{p_{\theta,t}}\right) \|\boldsymbol{s}_{p_{\theta,t}}(\boldsymbol{x}_t) - \boldsymbol{s}_{q_t}(\boldsymbol{x}_t)\|_2^2\right]\right\}$$

$$= -\frac{1}{2}g^2(t)\frac{\partial}{\partial \theta} \left\{\mathbb{E}_{p_{\theta,t}} \left[\mathrm{SG}\left(\mathcal{C}_1\left(\frac{q_t}{p_{\theta,t}}\right)\right) 2\left(\boldsymbol{s}_{q_t}(\boldsymbol{x}_t) - \boldsymbol{s}_{p_{sg[\theta],t}}(\boldsymbol{x}_t)\right) \left(\boldsymbol{s}_{p_{sg[\theta],t}}(\boldsymbol{x}_t) - \nabla_{\boldsymbol{x}_t} \log q_t(\boldsymbol{x}_t \mid \boldsymbol{x}_0)\right)\right]\right\}$$

$$- \frac{1}{2}g^2(t)\frac{\partial}{\partial \theta} \left\{\mathbb{E}_{p_{\theta,t}} \left[\mathrm{SG}\left(\mathcal{C}_2\left(\frac{q_t}{p_{\theta,t}}\right)\right) \left(\boldsymbol{s}_{q_t}(\boldsymbol{x}_t) - \boldsymbol{s}_{p_{\theta,t}}(\boldsymbol{x}_t)\right) \|\boldsymbol{s}_{q_t}(\boldsymbol{x}_t) - \boldsymbol{s}_{p_{\theta,t}}(\boldsymbol{x}_t)\|_2^2\right) \boldsymbol{x}_t\right]\right\} \tag{B.48}$$

where SG donates stop gradient operator, and the curvature coupling coefficient $\mathcal{C}(r)$ are defined as:

$$\mathcal{C}_1(r) := r^3 f'''(r), \quad \mathcal{C}_2(r) := 2r^2 f''(r) + 4r^3 f'''(r) + r^4 f''''(r), \quad r := \frac{q_t(\boldsymbol{x})}{p_{\theta,t}(\boldsymbol{x})} \tag{B.49}$$

$\square$

## B.3 Density Ratio Representation

**Theorem B.1** (Density Ratio Representation). *For adversarial discriminator conditioned on the timestep $t$ $D$: $\mathcal{X} \times [0, T] \to [0, 1]$ satisfying:*

$$D^* = arg \min_D \ \mathbb{E}_{\boldsymbol{x} \sim q_{\text{data}}}[-\log D(\boldsymbol{x}, t)] + \mathbb{E}_{\boldsymbol{x} \sim p_g}[-\log(1 - D(\boldsymbol{x}, t))], \qquad (B.50)$$

*The density ratio admits the variational representation:*

$$\frac{q_t(\boldsymbol{x})}{p_{\theta,t}(\boldsymbol{x})} = \frac{D^*(\boldsymbol{x}, t)}{1 - D^*(\boldsymbol{x}, t)}. \qquad (B.51)$$

*Proof of Theorem B.1.* Firstly, we calculate the optimal discriminator:

**Lemma B.3** (Optimal Discriminator Characterization). *For measurable functions $D : \mathcal{X} \times [0, T] \to [0, 1]$, the minimizer of:*

$$\mathcal{J}(D) = \mathbb{E}_{\boldsymbol{x} \sim q_t}[-\log D(\boldsymbol{x}, t)] + \mathbb{E}_{\boldsymbol{x} \sim p_{\theta,t}}[-\log(1 - D(\boldsymbol{x}, t))] \qquad (B.52)$$

*satisfies the first-order optimality condition:*

$$\left. \frac{\delta \mathcal{J}}{\delta D} \right|_{D=D^*} = -\frac{q_t(\boldsymbol{x})}{D^*(\boldsymbol{x}, t)} + \frac{p_{\theta,t}(\boldsymbol{x})}{1 - D^*(\boldsymbol{x}, t)} = 0. \qquad (B.53)$$

Solving Lemma B.3's optimality condition yields:

$$D^*(\boldsymbol{x}, t) = \frac{q_t(\boldsymbol{x})}{q_t(\boldsymbol{x}) + p_{\theta,t}(\boldsymbol{x})} \qquad (B.54)$$

Through algebraic transformation, we have:

$$\frac{q_t(\boldsymbol{x})}{p_{\theta,t}(\boldsymbol{x})} = \frac{D^*(\boldsymbol{x}, t)}{1 - D^*(\boldsymbol{x}, t)}. \qquad (B.55)$$

$\square$

## B.4 Proof of Corollary 3.4

*Proof of Corollary3.4.* Using Theorem3.1, assuming some mild assumptions on the growth of $\log q_t$ and $\log p_t$ at infinity, we have:

$$\mathcal{D}_f(q_0 || p_\theta) = \int_0^T \frac{1}{2} g^2(t) \mathbb{E}_{p_\theta} \left[ \left( \frac{q_t}{p_{\theta,t}} \right)^2 f'' \left( \frac{q_t}{p_{\theta,t}} \right) \|\nabla \log p_{\theta,t} - \nabla \log q_t\|_2^2 \right] dt. \qquad (B.56)$$

We also have the differential form of this formula:

$$\frac{d}{dt} \mathcal{D}_f(q_t || p_{\theta,t}) = -\frac{1}{2} g^2(t) \mathbb{E}_{p_{\theta,t}} \left[ \left( \frac{q_t}{p_{\theta,t}} \right)^2 f'' \left( \frac{q_t}{p_{\theta,t}} \right) \|\nabla \log p_{\theta,t} - \nabla \log q_t\|_2^2 \right]. \qquad (B.57)$$

We can re-weight Eq. B.56 for arbitrary weightings, where $W(t)$ is selected in our case. The re-weighted version of the RHS of Eq. B.56 can be written as:

$$\int_0^T \frac{1}{2} g^2(t) W(t) \mathbb{E}_{p_{\theta,t}} \left[ \left( \frac{q_t}{p_{\theta,t}} \right)^2 f'' \left( \frac{q_t}{p_{\theta,t}} \right) \|\nabla \log p_{\theta,t} - \nabla \log q_t\|_2^2 \right] dt. \qquad (B.58)$$

$$= \int_0^T -W(t) \frac{d}{dt} \mathcal{D}_f(q_t || p_{\theta,t}) dt. \qquad (B.59)$$

$$= -W(t) \mathcal{D}_f(q_t || p_{\theta,t}) \Big|_0^T + \int_0^T W'(t) \mathcal{D}_f(q_t || p_{\theta,t}) dt. \qquad (B.60)$$

$$= \int_0^T w(t) \mathcal{D}_f(q_t || p_{\theta,t}) dt. \qquad (B.61)$$

$\square$

# C  Detailed Analysis on $f$ Divergence

In this section, we provide several example divergences derived from our Uni-Instruct framework. Tab. 8 summarizes five types of divergence.

| Divergence | $f(r)$ | $\mathcal{C}_1(r)$ | $\mathcal{C}_2(r)$ | Mode-Seeking? |
|:---:|:---:|:---:|:---:|:---:|
| FKL | $r \log r$ | $-r$ | $0$ | - |
| RKL | $-\log r$ | $-1$ | $0$ | $\checkmark$ |
| JKL | $(r-1)\log r$ | $-r-1$ | $0$ | - |
| $\chi^2$ | $(r-1)^2$ | $0$ | $4r^2$ | - |
| JS | $r\log r - (r+1)\log\left(r+\frac{1}{2}\right)$ | $-\frac{r(2r+1)}{(r+1)^2}$ | $-\frac{2r^2}{(r+1)^3}$ | $\checkmark$ |

Table 8: Comparison of different $f$-divergences as a function of the likelihood ratio $r := \frac{q_t(\boldsymbol{x})}{p_{\theta,t}(\boldsymbol{x})}$

**Mode Seeking vs. Mode Covering**   For arbitrary $f$ divergence $\mathcal{D}_f(q||p) = \int p(\boldsymbol{x})f\left(\frac{q(\boldsymbol{x})}{p(\boldsymbol{x})}\right)\mathrm{d}\boldsymbol{x}$, it can be classified into two categories based on its mode seeking behavior. Divergences that are mode-seeking tend to push the generative distribution $p_\theta$ toward reproducing only a subset of the modes of the data distribution $p$. This selectivity is problematic for generative modeling because it can cause missing modes and reduce sample diversity. Such mode collapse has been noted for the integral KL loss employed in Diff-Instruct and DMD [30, 62]. A convenient way to quantify mode-seeking behavior is to inspect the limit $\lim_{r\to\infty} f(r)/r$: the smaller this limit grows, the stronger the mode-seeking tendency. Both reverse KL and Jensen–Shannon (JS) divergences have a finite value for this limit. By contrast, forward KL, Jeffrey KL, and $\chi^2$ yield an infinite limit, reflecting its well-known mode-covering nature, which tends to recover the entire data distribution $q$. In practice, we observed that mode covering divergences such as forward-KL and Jeffrey-KL achieves a lower FID score.

**Grad(SIM) vs. Grad(DI)**   Another way to inspect different $f$ divergence is checking the gradient expression. It is worth mentioning that the gradient expression of Uni-Instruct is composed of Grad(SIM) and Grad(DI) (Eq. 3.6). For KL divergence (reverse, forward, Jeffrey), $\mathcal{C}_2(r) = 0$ and the gradient is only contributed by Grad(SIM). On contrary, when selecting $\chi^2$ divergence, $\mathcal{C}_2(r) = 0$ and the gradient is only contributed by Grad(DI). The gradient expression of Jensen-Shannon (JS) is a combination of both.

**Training Stability**   However, during training we often observe training instability in Jensen-Shannon divergence and $\chi^2$ divergence, due to the complex expression of $\mathcal{C}_1(r)$ and $\mathcal{C}_2(r)$, which will result in higher FID score (Tab. 5). Tricks such as normalizing the weighting function or implementing the discriminator on the teacher model [59] can be applied to stabilize training. We leave this part to future work.

# D  Unified Distillation Loss

In this section, we discuss how Uni-Instruct unifies previous diffusion distillation methods through recovering previous methods into a special case of Uni-Instruct. We summarize the connections in Tab. 1.

## D.1  One Step Diffusion Model Distillation

From Section 3.3 and Corollary 3.4, we have demonstrated that integral KL-based divergence minimization can be treated as Uni-Instruct with special weighting. More surprisingly, we found that if we choose $\chi^2$-divergence in Uni-Instruct, the weighting of SIM becomes $0$ and the remaining gradient is only contributed by Diff-Instruct, as is shown in Tab. 8 and the third column of Tab. 1. In this way, Uni-Instruct can unify the first line of work: Diff-Instruct [30] is Uni-Instruct with $\chi^2$-divergence. DMD [62] added extra regression loss contributed by pre-sampled paired images, while DMD2 [61] added an adversarial loss. SwiftBrush [16] applied the same loss on text-to-image

generation. $f$-distill [59] can be seen as Uni-Instruct with manually selected weighting, and has a gradient expression of ($\chi^2$) divergence in Uni-Instruct.

Moreover, in Sec. 3.3, we demonstrate that leveraging the connection between KL divergence and score-based divergence, score matching can be interpreted as minimizing single-step KL divergence. Thus, selecting reverse-KL (RKL) divergence in Uni-Instruct, we can recover score-based divergence, as shown in the third column of Tab. 1. In this way, SIM [31] and SiD [69] minimize Uni-Instruct loss with RKL. Additional adversarial loss is added in SiDA[68], while text-to-image distillation is applied in SID-LSG[67], both under the same Uni-Instruct(RKL) setting. Though our experiments on benchmark datasets have already demonstrated the superior performance of Uni-Instruct on distilling a one-step diffusion model (Sec. 4). We believe Uni-Instruct can be further applied to large-scale datasets and text-to-image diffusion models. We leave that to future work.

## D.2 Text-to-3D Generation with Diffusion Distillation

DreamFusion [41] and ProlificDreamer [55] propose to leverage text-to-image diffusion models to distill neural radiance fields (NeRF) [36], enabling efficient text-to-3D generation from a fixed text prompt. DreamFusion utilizes a pretrained text-to-image diffusion model to guide the optimization of a NeRF network by performing score-distillation sampling (SDS). This method minimizes KL divergence that aligns the rendered images from NeRF with the guidance from a pretrained diffusion model.

ProlificDreamer further advances this concept by introducing variational distillation, which involves training an extra student network to stabilize and enhance the distillation process. Specifically, denote $p_\theta(\boldsymbol{x}|c, y)$ as the implicit distribution of the rendered image $\boldsymbol{x} := \boldsymbol{g}(\theta, c)$ given the camera $c$ with the rendering function $\boldsymbol{g}(\cdot, c)$, while $q_0(\boldsymbol{x}|y^c)$ as the distribution modeled by the pretrained text-to-image diffusion model with the view-dependent prompt $y^c$. ProlificDreamer approximates the intractable implicit distribution posterior distribution $p_\theta(\boldsymbol{x}|c, y)$ by minimizing the integral KL divergence between the diffusion-guided posterior and the implicit distribution rendered by NeRF:

$$\mathcal{D}_{\text{IKL}}(p_\theta(\boldsymbol{x}|c, y)||q_0(\boldsymbol{x}|y^c)) := \int_0^T w(t)\mathbb{E}p_{\theta,t}(\boldsymbol{x}_t|c, y)\left[\log\frac{p_{\theta,t}(\boldsymbol{x}_t|c, y)}{q_t(\boldsymbol{x}_t|y^c)}\right]\mathrm{d}t. \tag{D.1}$$

Utilizing Corollary 3.4, we observe that by choosing suitable weighting functions $W(t)$, the integral KL divergence used by ProlificDreamer corresponds to the reverse KL (RKL) version of Uni-Instruct:

$$\int_0^T w(t)\mathcal{D}_{\text{KL}}(p_{\theta,t}(\boldsymbol{x}_t|c, y)||q_t(\boldsymbol{x}_t))\mathrm{d}t = \int_0^T \frac{1}{2}g^2(t)W(t)\mathbb{E}_{p_{\theta,t}}\left[|\boldsymbol{s}_{p_{\theta,t}}(\boldsymbol{x}) - \boldsymbol{s}_{q_t}(\boldsymbol{x})|_2^2\right]\mathrm{d}t, \tag{D.2}$$

ignoring $W(t)$ becomes the RKL loss function we applied in our experiments.

Moreover, the gradient expression of DreamFusion and ProlificDreamer can be seamlessly unified under the Uni-Instruct framework, specifically aligning with the $\chi^2$ divergence case of Uni-Instruct (third column of Tab. 1). Our experiments indicate that employing Uni-Instruct with KL-based divergence in the text-to-3D setting slightly improves the quality of generated 3D objects (App. E).

## D.3 Solving Inverse Problems with Diffusion Distillation

To solve a general noisy inverse problem, which seeks to find $\boldsymbol{x}$ from a corrupted observation:

$$\boldsymbol{y} = h(\boldsymbol{x}) + v, v \sim \mathcal{N}(0, \sigma_v^2\boldsymbol{I}) \tag{D.3}$$

where the forward model $h$ is known, we aims to compute the posterior $p(\boldsymbol{x}|\boldsymbol{y})$ to recover underlying signals $\boldsymbol{x}$ from its observation $\boldsymbol{y}$. The intractable posterior $p(\boldsymbol{x}|\boldsymbol{y})$ can be approximated by $q(\boldsymbol{x}|\boldsymbol{y})$ through variational inference, where $q := \mathcal{N}(\mu, \sigma^2\boldsymbol{I})$ is the variational distribution. Starting from minimizing the KL divergence between these two distributions, we have:

$$\mathcal{D}_{\text{KL}}(p_\theta(\boldsymbol{x}|\boldsymbol{y})||p(\boldsymbol{x}|\boldsymbol{y})) = -\mathbb{E}_{q(\boldsymbol{x}|\boldsymbol{y})}\left[\log p(\boldsymbol{y}|\boldsymbol{x})\right] + \mathcal{D}_{\text{KL}}\left(p_\theta(\boldsymbol{x}|\boldsymbol{y})||q(\boldsymbol{x})\right) + \log p(\boldsymbol{y}), \tag{D.4}$$

where the first term is tractable base on the forward model of the inverse problem and the third term is irrelevant to the optimization problem. RedDiff [34] proposed to estimate the second term with diffusion distillation. Specifically, they expand the KL term with integral KL through manually adding

time weighting $w(t)$: $\mathcal{D}_{\text{IKL}}(p_\theta(\boldsymbol{x}|\boldsymbol{y})\|q_0(\boldsymbol{x})) := \int_{t=0}^{T} w(t)\mathbb{E}_{p_{\theta,t}(\boldsymbol{x}_t|\boldsymbol{y})}\left\{\log\frac{p_{\theta,t}(\boldsymbol{x}_t|\boldsymbol{y})}{q_t(\boldsymbol{x}_t)}\right\}\mathrm{d}t$. Using Corollary 3.4, choosing $W(t) = \int w(t)dt + C, W(0) = 0$, we can recover the RKL version of Uni-Instruct:

$$\int_0^T w(t)\mathcal{D}_{\text{KL}}(p_{\theta,t}(\boldsymbol{x}_t|\boldsymbol{y})\|q_t(\boldsymbol{x}_t))\mathrm{d}t = \int_0^T \frac{1}{2}g^2(t)W(t)\mathbb{E}_{p_{\theta,t}}\left[\|\boldsymbol{s}_{p_{\theta,t}}(\boldsymbol{x}|\boldsymbol{y}) - \boldsymbol{s}_{q_t}(\boldsymbol{x})\|_2^2\right]\mathrm{d}t. \quad \text{(D.5)}$$

### D.4 Human Preference Aligned Diffusion Models

Reinforcement learning from human feedback [39, 7] (RLHF) is proposed to incorporate human feedback knowledge to improve model performance. The RLHF method trains the model to maximize the human reward with a Kullback-Leibler divergence regularization, which is equivalent to minimizing:

$$\mathcal{L}(\theta) = \mathbb{E}_{\boldsymbol{x}\sim p_\theta(\boldsymbol{x})}\left[-r(\boldsymbol{x})\right] + \beta\,\mathcal{D}_{\text{KL}}\left(p_\theta(\boldsymbol{x})\|q_{\text{ref}}(\boldsymbol{x})\right) \quad \text{(D.6)}$$

The KL divergence regularization term penalizes the distance between the optimized model and the reference model to prevent it from diverging, while the reward term encourages the model to generate outputs with high human rewards. After the RLHF finetuning process, the model will be aligned with human preferences.

The KL penalty in Eq. D.6 can be performed with diffusion distillation when aligning the diffusion model with human preference. DI++ [29] propose to penalize the second term with IKL, which minimizes the KL divergence along the diffusion forward process:

$$\mathcal{L}(\theta) = \mathbb{E}_{\substack{z\sim p_z,\,\boldsymbol{x}_0=g_\theta(z)\\x_t|x_0\sim p(x_t|x_0)}}\left[-r(\boldsymbol{x}_0)\right] + \beta\int_0^T w(t)\,\mathcal{D}_{\text{KL}}\left(p_\theta(\boldsymbol{x}_t)\|q_{\text{ref}}(\boldsymbol{x}_t)\right)\,dt \quad \text{(D.7)}$$

Alternatively, DI* [32] replaces the integral KL divergence with score-based divergence:

$$\mathcal{L}(\theta) = \mathbb{E}_{\boldsymbol{x}_0\sim p_\theta(x_0)}\left[r(\boldsymbol{x}_0)\right] + \beta\int_0^T \frac{1}{2}g^2(t)\,\|\boldsymbol{s}_{p_{\theta,t}}(\boldsymbol{x}_t) - \boldsymbol{s}_{q_t}(\boldsymbol{x}_t)\|_2^2\,dt \quad \text{(D.8)}$$

Leveraging Corollary 3.4, the integral KL divergence in Eq. D.7 is a weighted version of KL divergence. Choosing $W(t) = \int w(t)dt + C, W(0) = 0$, we have:

$$\int_0^T w(t)\mathcal{D}_{\text{KL}}(q_0\|p_{\theta,t})\mathrm{d}t = \int_0^T \frac{1}{2}g^2(t)W(t)\mathbb{E}_{p_{\theta,t}}\left[\|\boldsymbol{s}_{p_{\theta,t}}(\boldsymbol{x}_t) - \boldsymbol{s}_{q_t}(\boldsymbol{x}_t)\|_2^2\right]\mathrm{d}t. \quad \text{(D.9)}$$

Moreover, the score based divergence in Eq. D.8 is minimizing KL divergence $D_{\text{KL}}(p_\theta(\boldsymbol{x})\|q_{\text{ref}}(\boldsymbol{x}))$, based on Theorem 3.1, which recovers the RKL version of Uni-Instruct.

The gradient of DI++ [29] and DI* [32] takes the form of DI [30] and SIM [31], which correspond to $\chi^2$ and RKL divergence separately (third column of Tab. 1).

## E  Details of 3D Experiments

**Experiment Settings**  In this section, we elaborate on the implementation details of Uni-Instruct on text-to-3D generation. We re-implement the code base of ProlificDreamer [55] by adding an extra discriminator head to the output of the stable diffusion Unet's encoder. We apply forward-KL and reverse-KL to Uni-Instruct and train the NeRF model. To further demonstrate the visual quality, we transform the NeRF to mesh with the three-stage refinement scheme proposed by ProlificDreamer: (1) Stage one, we use Uni-Instruct guidance to train the NeRF model for 300~400 epochs, based on the model's performance on different text prompts. (2) Stage 2, we obtain the mesh representation from the NeRF model and use the SDS loss to fine-tune the object's geometry appearance for 150 epochs. (3) Stage 3: We add more vivid texture to the object through further finetuning with Uni-Instruct guidance for an additional 150 epochs. Additionally, we enhance the object's appearance with a human-aligned loss provided by a reward model. Algorithm 2 shows how to distill a 3D NeRF model.

**Performance Evaluations**  Fig. 4 shows the objects produced by the mesh backbone. Uni-Instruct produces more diverse results compared to ProlificDreamer and DreamFusion. Fig. 5 demonstrates more objects trained with the NeRF backbone. Tab. 9 shows the numerical results. Our method slightly outperforms the baseline methods.

**Algorithm 2:** Uni-Instruct for Text-to-3D Generation

**Input:** pre-trained DM $s_{q_t}$, generator $g_\theta$, fake score network $s_\psi$, discriminator $D_\lambda$, divergence $f$, GAN weight $w_{\text{GAN}}$, diffusion timesteps weighting $w(t)$.

1: **while** not converge **do**
2:     Sample camera view $c$ and random noises: $\epsilon \sim \mathcal{N}(0, I)$
3:     Render fake images from NeRF: $\boldsymbol{x} = \boldsymbol{g}(\theta, c)$
4:     Sample real images and random noises: $\boldsymbol{x}_{\text{real}} \sim p_{\text{data}}, \epsilon \sim \mathcal{N}(0, I)$
5:     Update $D_\lambda$ with discriminator loss:
    $\mathcal{L}_D = -\mathbb{E}_{\boldsymbol{x}_{\text{real}}}[\log D_\lambda(\boldsymbol{x}_{\text{real}})] - \mathbb{E}_{\boldsymbol{x}_{\text{fake}}}[\log(1 - D_\lambda(\boldsymbol{x}_{\text{fake}}))]$
6:     Compute diffusion guidance:
    $\mathcal{L}_{\text{diffusion}} = \int_0^T w(t)\, \mathbb{E}_{\boldsymbol{x}_t|\boldsymbol{x} \sim p_{\theta,t}(\boldsymbol{x}_t|\boldsymbol{x})} \left\| s_{p_{\theta,t}}(\boldsymbol{x}_t) - s_{q_t}(\boldsymbol{x}_t) \right\|_2^2 \mathrm{d}t$
7:     Compute Uni-Instruct loss: $\mathcal{L}_{\text{Uni}}$ (Equation 3.5)
8:     Update $\theta$ with $\mathcal{L}_{\text{Uni}}$.
9: **end while**
10: **return** $g_\theta$

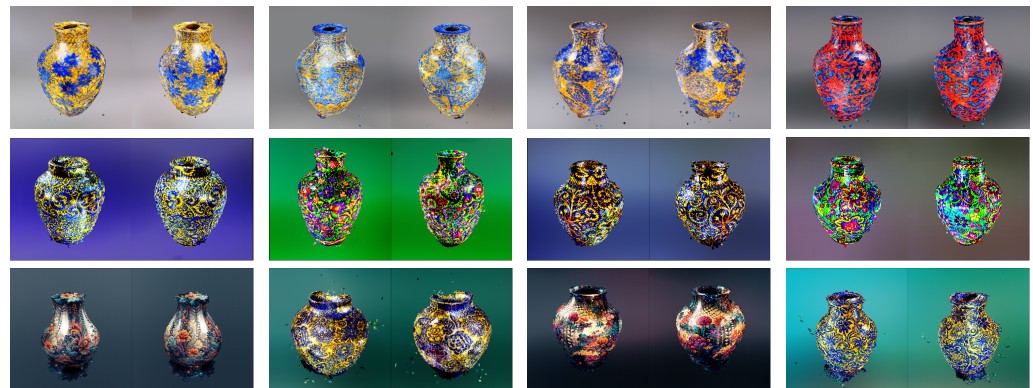

Figure 4: **Prompt**: A refined vase with artistic patterns. *From top to bottom :* ProlificDreamer, Uni-Instruct (Forward-KL), Uni-Instruct (Reverse-KL). Our vase demonstrates more diverse shapes as well as realistic patterns.

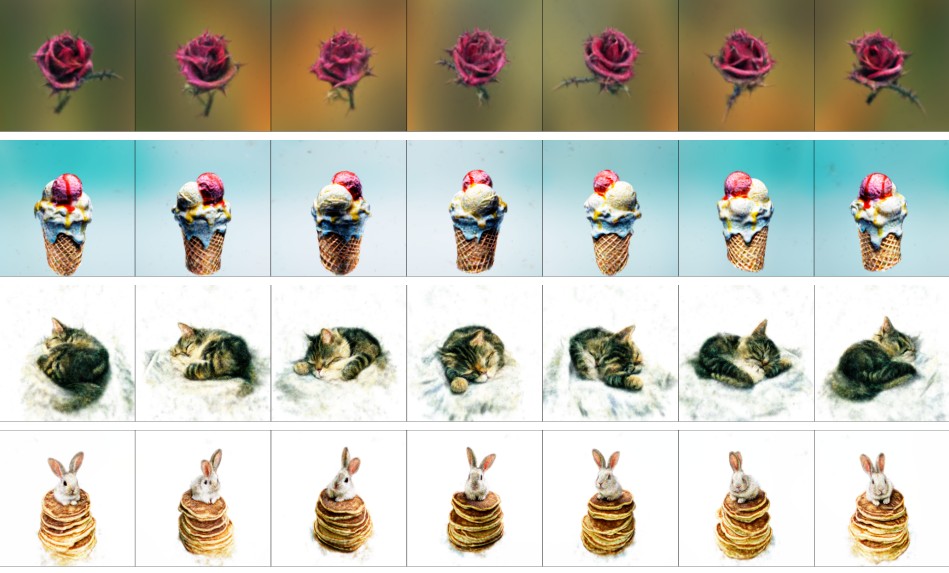

Figure 5: Results generated from our NeRF backbone. **Prompts (From top to buttom)**: "A thorny rose.", "A high-quality photo of an ice cream sundae.", "A sleeping cat.", "A baby bunny sitting on top of a stack of pancakes."

Table 9: Comparison of different methods on Mesh and NeRF backbones. The prompt is: "A refined vase with artistic patterns."

| Method | NeRF | | Mesh | |
|---|---|---|---|---|
| | 3D-Aes Score↑ | 3D-CLIP↑ | 3D-Aes Score↑ | 3D-CLIP↑ |
| DreamFusion [41] | 1.07 | 27.79 | - | - |
| Fantasia3D [5] | - | - | 2.76 | 30.96 |
| ProlificDreamer [55] | 2.15 | 30.97 | 4.91 | 31.92 |
| **Uni-Instruct (Forward-KL)** | 2.46 | 31.35 | 4.83 | 31.74 |
| **Uni-Instruct (Reverse-KL)** | **4.45** | **33.94** | **7.54** | **34.56** |

## F  Limitaions

Training an additional discriminator to estimate the density ratio brings extra computational costs and may lead to unstable training. For instance, we found that the output of a 3D object trained with Uni-Instruct forward KL is more foggy than reverse KL, which doesn't require an extra discriminator. Additionally, Uni-Instruct suffers from slow convergence: Training Uni-Instruct on both 2D distillation and text-to-3D tasks takes twice as long as training DMD and ProlificDreamer on their respective tasks. Moreover, Uni-Instruct may result in bad performance with an improper choice of $f$, as the gradient formula in Eq. 3.3 requires the fourth derivative of function $f$, which will add complexity to the gradient formula. Therefore, Uni-Instruct is not as straightforward as some simpler existing methods like Diff-Instruct. We hope to develop more stable training techniques in future work.

## G  Additional Results

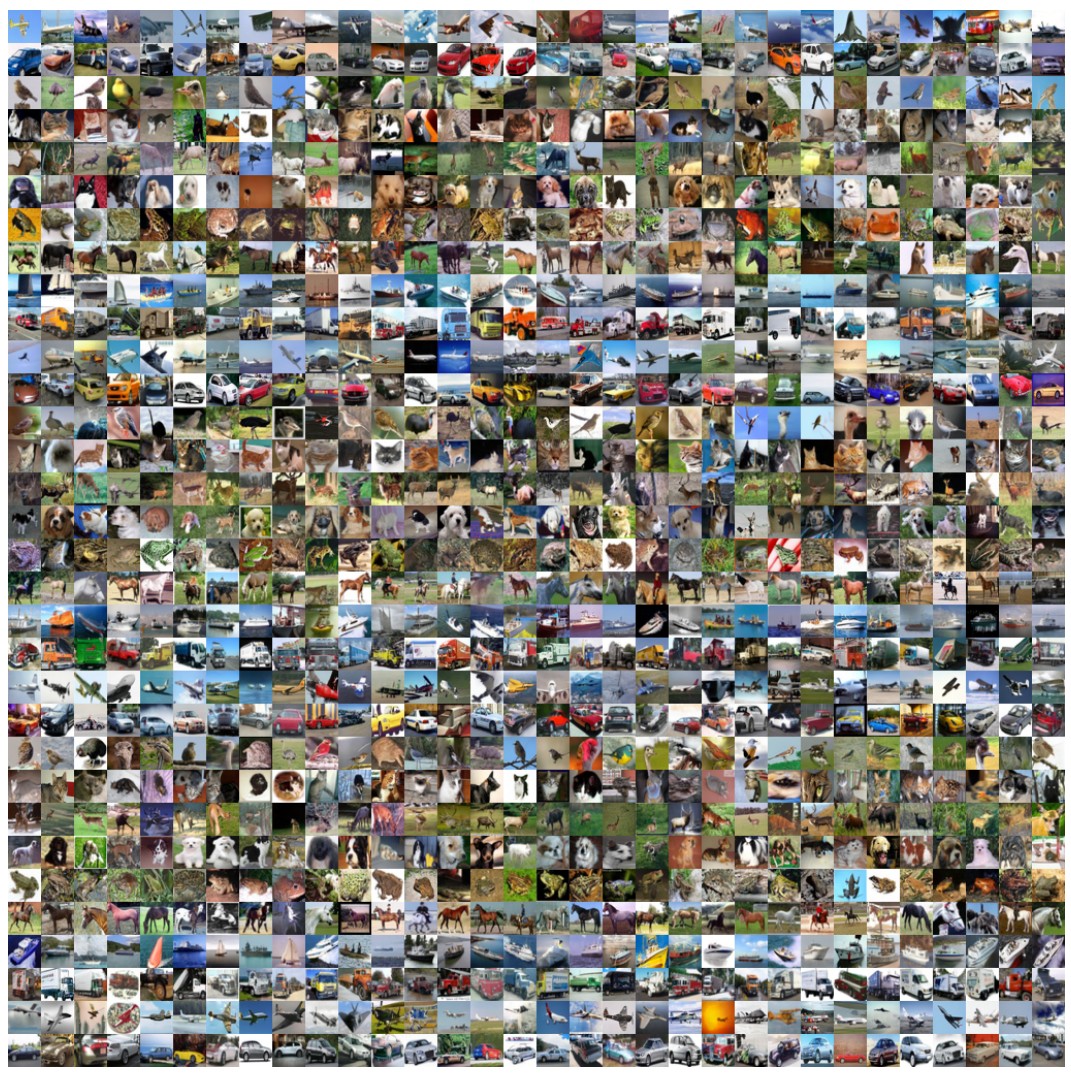

Figure 6: Forward-KL CIFAR10 conditional generation.

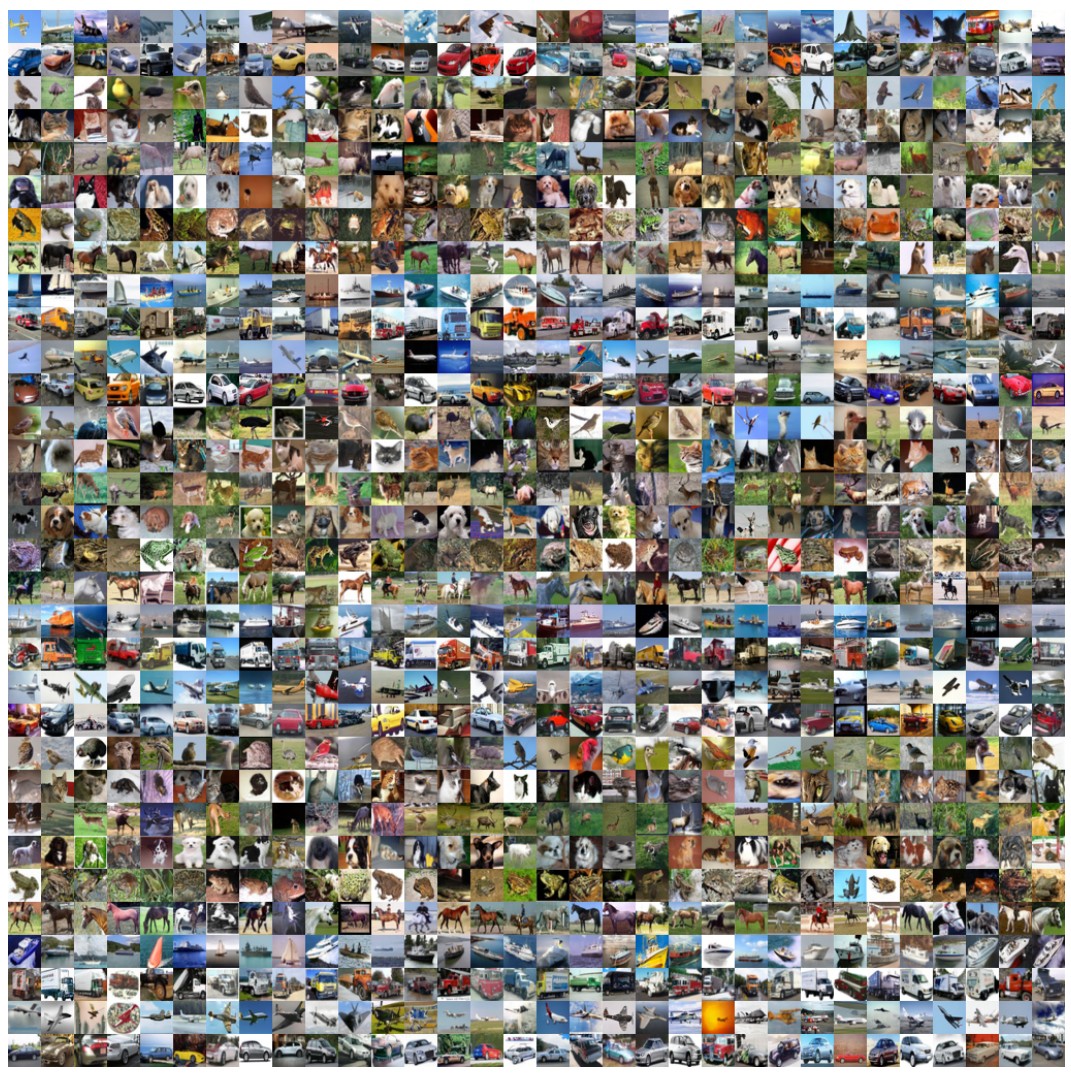

Figure 7: Jeffrey-KL CIFAR10 conditional generation.

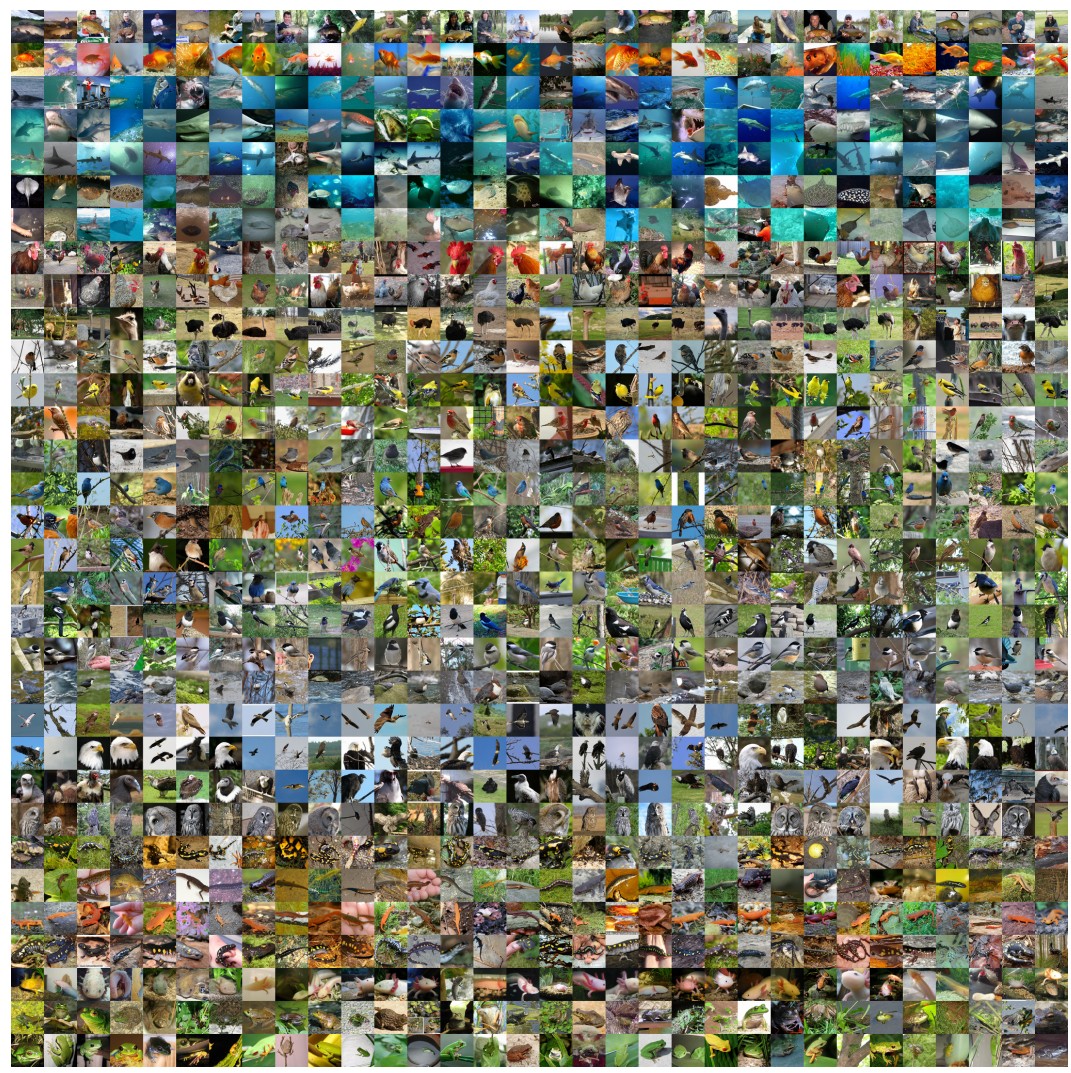

Figure 8: Forward-KL ImageNet64 conditional generation.

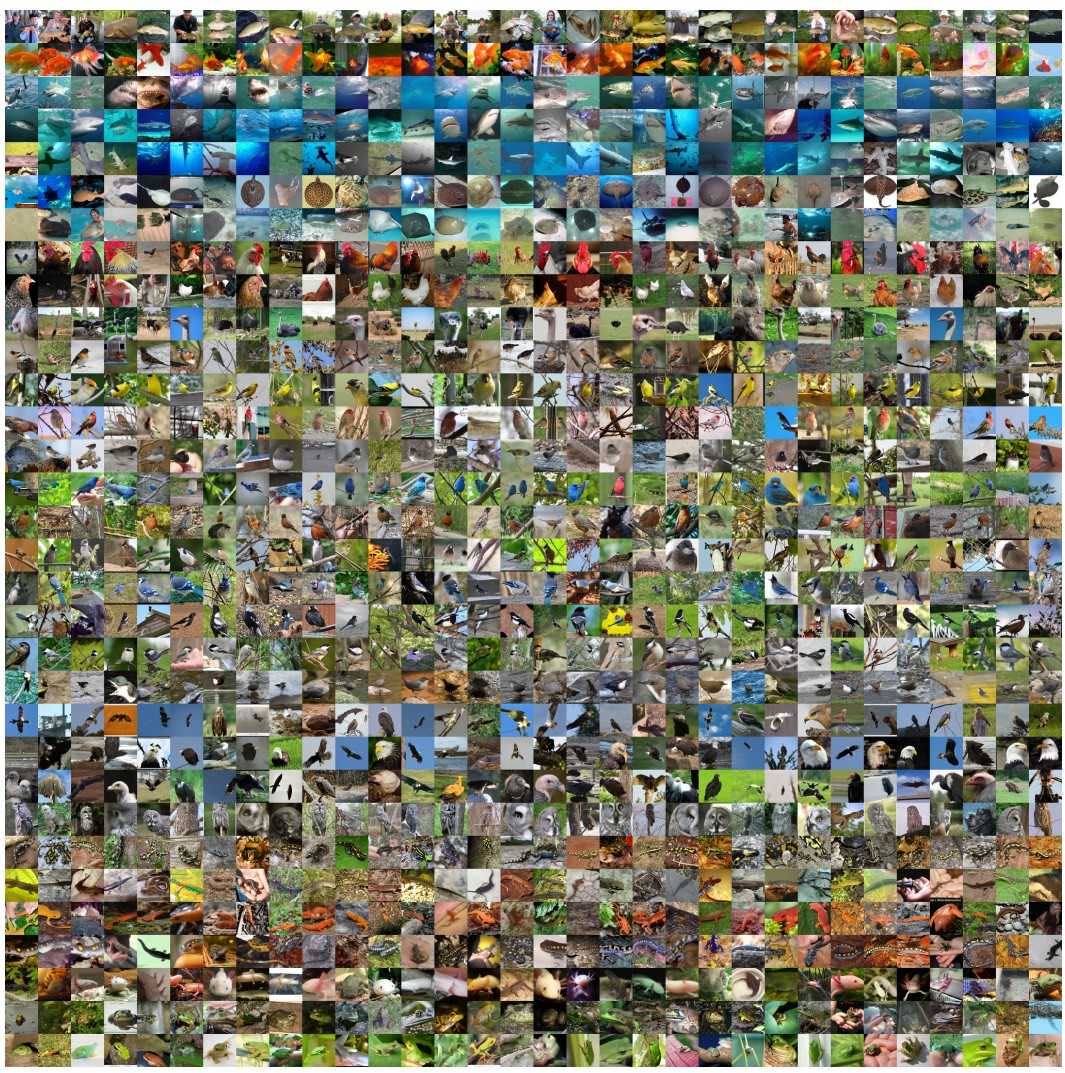

Figure 9: Jeffrey-KL ImageNet64 conditional generation.

