# OpenReview forum: "Uni-Instruct: One-step Diffusion Model through Unified Diffusion Divergence Instruction"
_NeurIPS.cc/2025/Conference — NeurIPS 2025 poster_

### Official Review · Reviewer_cqxy · 2025-06-29

**Clarity:** 3
**Significance:** 3
**Originality:** 3
**Rating:** 4
**Confidence:** 4

**Summary:**

This paper aims to propose a unified framework for existing score distillation techniques. By leveraging the observation that the Fisher divergence can also be viewed as the KL divergence between the path measure under the diffusion SDE case, they propose a time-weighted f-divergence between the diffusion SDE as their unified framework, which allows a principled way to combine the (weighted) Diff-Instruct and SiD/SIM loss for distilling from a diffusion model. The experimental results demonstrate the effectiveness of both the image generation benchmark and text-to-3D generation.

**Questions:**

* Given the mode-seeking / coverage behavior of the difference divergence between two distributions is well-established, how does the same intuition apply to the integration of them? Is there more explicit evidence to support the same mode-coverage / seeking behavior applies in this case, either mathematically or empirically?

**Ethical Concerns:**

["NO or VERY MINOR ethics concerns only"]

**Final Justification:**

The main concern regarding clarity, especially the crucial hidden details of the methodology part, is properly addressed.

I am happy to give the manuscript "Weak Accept" for now since I still believe this manuscript needs a certain level of revision, but the paper cannot be revised during the rebuttal period.

**Limitations:**

yes

**Quality:**

3

**Strengths And Weaknesses:**

## Strengths
* The empirical performance in terms of FID is strong, setting a new record on the ImageNet $64 \times 64$ benchmark.
* The proposed unified framework unlocks a principled perspective to combine and leverage (i.e., weighting) the established score distillation techniques.
* Unveiling the connection between the weighted integral divergence and divergence between path measures is a nice contribution to me.

## Weaknesses

### Major
My major concern for this manuscript is about the writing clarity, as well as the lack of support for their claim regarding the method and experiments.

* A relevant work [1] should be discussed, which also illustrates the connection between the KL divergence between the path measure and the Fisher divergence.

* I have some confusion regarding Theorem 3.2:
   * For the RKL case, it is equivalent to the Fisher divergence (up to some weighting factor), and previous works [1,2] show that it has another term than the $\mathcal L_{\mathrm{SIM}}$ here. However, Theorem 3.1 suggests that the RKL case is equivalent to a weighted version of the original $\mathcal L_{\mathrm{SIM}}$ in Eqn. (2.3). How do these different results reconcile?
   * The derivation of Theorem 3.2 is somewhat hard for me to follow. For example, in the calculation of $A$ involves the inversion of  $q_t / p_{\theta, t}$ inside $f^{\prime\prime}$, which is unclear to me. Besides, it seems the derivation suggests the proposed diffusion expansion of $f$-Divergence is symmetric w.r.t. the gradient operation by looking at Eqn. (A.22) and Eqn. (A.43), which needs more justification than provided.

(I might have missed something here, but I think explicitly clarifying the connection of the unified framework with previous works would be a good contribution to this line of research. I am happy to raise my rating if this point clears up.)

* Some of the other presentations/claim confuses me:
  * To me, it is somewhat confusing to refer to the two terms in Eqn. (3.6) as $\mathcal L_{\text{SIM}}$ and $\mathcal L_{\mathrm{DI}}$ as the weighting here depends on the density ratio and the score difference. This is intrinsically different from the time-weighted SIM and DI objective via $\omega(t)$.
  * From line 232, the claim about SIM needs more justification. The $\pi_t$ in SIM is chosen as the $p_{\mathrm{sg}[\theta]}(\boldsymbol x_t)$, which generally is not the case for the reverse KL divergence between path measures, i.e., $\frac{1}{2}g^2(t) \int \mathbb E_{p_{\theta, t}} [\| \boldsymbol s_{p_{\theta, t}}(\boldsymbol x_t) - \boldsymbol s_{q_t}(\boldsymbol x_t) \|]$.

* Some clarity issues:
 1. In line 137, the $\pi_t$ in the general score-divergence is undefined.
 1. In lines 298-300, the $\mathrm{Grad(SiD)}$ is inconsistent with the previous usage of SIM.

* Several claims regarding the empirical performance need to be reexamined:
  * In lines 50-51, the description of surprising generation performance and sub-optimal fidelity forms a contradiction to me..
  * In line 283, FID is not necessarily a gold metric for diversity. The author is encouraged to consider other metrics (i.e., the recall scores used in [3, 4]) to demonstrate the claimed effectiveness via the combination of mode-seeking and mode-coverage behavior of the proposed method.

[1] Gushchin, Nikita, et al. [Inverse Bridge Matching Distillation](https://openreview.net/forum?id=UCJSF6Vt0C). (ICML 2025)

[2] Huang, Zemin, et al. [Flow Generator Matching](https://arxiv.org/abs/2410.19310). (2024)

[3] Lu, Cheng, and Yang Song. [Simplifying, Stabilizing and Scaling Continuous-Time Consistency Models](https://openreview.net/pdf?id=LyJi5ugyJx). (ICLR 2025)

[4] Sabour, Amirmojtaba, et al. [Align Your Flow: Scaling Continuous-Time Flow Map Distillation](https://arxiv.org/abs/2506.14603). (2025)

### Minor
* I didn't quite get the point of highlighting some of the sentences with blue color. For example, why does the author think lines 208-209 should be highlighted?
* The readability of Fig. 1 (right) is not great, and the meaning of the bigger size of the Uni-Insturct is unclear.
* The format for plugging the reference number is not unified, for example, whether to use Diff-Instruct[31] or Diff-Instruct [31].
* The inconsistency in math notations, a non-exhaustive example is:
   * In line 141, the $\mathcal L_{SIM}$ should be like $\mathcal L_{\mathrm{SIM}}$
   * In line 541, $p_{\theta, t}$ or $p_t$ in Eqn. (A.30) and Eqn. (A.31)
   * Some glitches in the proof of Lemma A.2, i.e., $p_{\theta}, 0$ should be $p_{\theta, 0}$ from Eqn. (A.12)? and $x_t$ should be $\boldsymbol x_t$ in Eqn. (A.21)

---

> ### Author Rebuttal · Authors · 2025-07-31
>
> Thank you for your feedback. Below, we will address your concerns one by one:
>
> **Q1**: It would be good if the authors could discuss paper [1].
>
> **A1**: Thank you for bringing paper [1] to our attention, which bridges the KL and Fisher divergence. However, paper [1] has an essentially different scope and focus from our Uni-Instruct. The contribution of Uni-Instruct lies in proposing a unified theoretical framework named Uni-Instruct together with a novel $f$-divergence expansion theorem, as well as its successful implementation in ImageNet$64\times64$ and text-to-3D generation. We will discuss the relationship between Uni-Instruct and [1] in the revised version.
>
> **Q2**: Could the authors clarify more on $\mathcal{L}\_{\text{FGM}}$ and $\mathcal{L}\_{\text{SIM}}$ regarding Theorem 3.2?
>
> **A2**:
> We appreciate your good intuition on $\mathcal{L}\_{\text{FGM}}$ and $\mathcal{L}\_{\text{SIM}}$. From a theoretical perspective, $\mathcal{L}\_{\text{FGM}}$, is a combination of $\mathcal{L}\_{\text{SIM}}$ and an additional term related to the gradient dependence of the generator distribution. In our theory part, we did not cut off the gradient dependence of the generated images, and the correct term should be FGM. However, prior work (such as SIM) has shown that under the assumption that the generated image distribution is independent of the gradient, this additional term does not influence the final distillation outcome. Therefore, to simplify the loss expression and analysis, we adopt $\mathcal{L}\_{\text{SIM}}$ instead of the full $\mathcal{L}\_{\text{FGM}}$ in our final loss expression.
>
> **Q3**: Could the authors clarify the proof of Theorem 3.2 step-by-step? Particularly the calculation of $A$ involves the inversion of $q_t/p_{\theta,t}$ inside $f''$. Also, is the proposed diffusion expansion of $f$-Divergence symmetric with respect to the gradient operation by looking at Eqn. (A.22) and Eqn. (A.43)?
>
> **A3**: We feel sorry for the confusion. We acknowledge that there is a non-technical Typo in Eqn. (A.31): the calculation of A (Eqn. (A.31), Eqn. (A.32), and Eqn. (A.33)) and Eqn. (A.43) mistakenly swapped $p$ and $q$. $p$ should be $q$, while $q$ should be $p_{\theta}$.
>
>
> Next, we would like to clarify that the proposed diffusion expansion of $f$-Divergence is not symmetric: $q$ is the distribution modeled by the pretrained diffusion, while $ p_\theta$ is the distribution modeled by the student generator. We sample from the generator to calculate the expectation. The correct version of Eqn. (A.43) should be the same as Eqn. (A.22) and the correct version of Eqn. (A.31), Eqn. (A.32), and Eqn. (A.33) should exchange $p$ and $q$. Thank you for your valuable feedback! We will correct our typo in the version.
>
> **Q4**: Could the authors discuss more on the weighting functions of Eqn. (3.6) as $\mathcal{L}\_{\text{SIM}}$ and $\mathcal{L}\_{\text{DI}}$.
>
> **A4**:  We agree that the Uni-Instruct involves density-ratio and score-related weighting functions. However, the density-ratio and score-terms inside the weighting function do not contribute to the parameter gradient of the generator; therefore can be safely viewed as a part of the weighting function. We are glad to discuss more.
>
> In the original DI and SIM paper, the authors built clear theories that led to time-dependent weighting functions without density-ratio and score terms. However, Fisher divergence (as well as general score-based divergence) can be viewed as a special case of the expanded $f$-divergence in our Theorem 3.2. With this observation, Uni-Instruct is built in a top-down manner, which starts from $f$-divergence expansion, and ends at two separated terms that have principal parts of DI and SIM, but with a density-ratio related weighting function, which is implemented by introducing a density-ratio estimator (GAN discriminator).
>
> **Q5**: Could the authors clarify more on the SIM loss: $\pi\_t$ in SIM is chosen as $p\_{\text{sg}[\theta]}(\boldsymbol{x}\_t)$, what is the relation between it and the reversed KL divergence?
>
> **A5**: In our theory part, we take the parameter dependence into account, and the original term is FGM. However, based on the theoretical part from SIM[2], cutting off the gradient dependence of the sampling distribution still leads to a valid divergence. So we use SIM in our final loss expression for simplicity. Sorry for the confusion, we will clarify this point in the revised version.
>
> For the reversed KL divergence, according to our $f$-divergence expansion theorem (reverse KL is a special case of $f$-div), the divergence could lead to an additional term. However, in our paper, we ignore the additional term from FGM for two reasons: (1) SIM shows better empirical performance than FGM, empirically indicating the additional term does not bring strong positive effects; (2) similar to SIM, since the expanded $f$-divergence in Theorem 3.2 still hold an squared of an L2 norm form, cutting off the parameter dependence in sampling distribution still lead to a valid divergence. However, we highly appreciate your wonderful intuition and would like to add a thorough discussion on parameter cutting-off in the revision.
>
>
> **Q6**: Clearity issues: the definition of $\pi\_t$. In lines 298-300, $\text{Grad(SiD)}$ is inconsistent with the previous usage of SIM.
>
> **A6**: We are sorry for the confusion. $\pi\_t$ is the distribution modeled by the one-step generator, with a stop gradient on the generated images: $p\_{\text{sg}[\theta]}(\boldsymbol{x}\_t)$. And we will correct $\text{Grad(SiD)}$ into $\text{Grad(SIM)}$ in the revision.
>
> **Q7**: Could the authors clarify more on the description of the surprising generation performance and sub-optimal fidelity in lines 50-51?
>
> **A7**: We feel sorry for the confusion. We want to express that when compared with KL-div based distillation methods, the score-based distillation shows better performance. However, score divergence-based distillation converges more slowly than KL-div, and may still have room for improved performance if we consider more general divergences such as expanded $f$-divergence. This belief motivates us to explore theoretically driven approaches to not only unify existing distillation methods but also seek better empirical performance.
>
>
> **Q8**: The author is encouraged to evaluate models under more metrics (i.e., the precision-recall scores) to see the pros and cons of the approach.
>
> **A8**: Thank you for your valuable suggestion. In the rebuttal phase, we evaluate the Uni-Instruct one-step model with other baseline models under 5 more metrics: $\text{FD}\_{\text{dino}}$, precision, recall, IS (Inception Score), and sFID. Due to space limits, please check **Table 1** and the analysis in **A3** of our response to **Reviewer TDUe**.
>
>
> **Q9**: Given the mode-seeking / coverage behavior of the difference divergence between two distributions is well-established, how does the same intuition apply to the integration of them? Is there more explicit evidence to support the same mode-coverage / seeking behavior applies in this case, either mathematically or empirically?
>
> **A9**: The $f$-divergence expansion serves as a general framework that encompasses SIM, SiD, and SiDA as special cases. In practice, we find that using the Jeffrey KL divergence significantly outperforms other instances of $f$-divergence. One possible reason is that the Jeffrey divergence combines both the forward and reverse KL, thereby balancing the mode-covering behavior of the forward KL with the mode-seeking tendency of the reverse KL, which leads to more stable training and better generation quality.
>
> **Dear reviewer. We appreciate your time and valuable suggestions. We hope our rebuttal has resolved all your concerns. If you still have remaining concerns, please do let us know. We are glad to provide more clarifications!**
>
> Best,
> Authors of submission # 3209
>
> [1] Gushchin et al. Inverse Bridge Matching Distillation. (ICML 2025)
>
> [2] Luo et al. One-Step Diffusion Distillation through Score Implicit Matching. (NeurIPS 2024)

---

> > ### Comment · Reviewer_cqxy · 2025-08-04
> >
> > Thank the authors for their rebuttal.
> >
> > I appreciate the clarification for answering my questions, as well as the extra empirical results. I think this paper has its merits and is making valid contributions.
> >
> > However, given that the paper cannot be updated during the author-reviewer discussion phase, and the fact that a major revision is required in the methodology part. Could the author kindly provide a more detailed formal draft / outline about the "theory part" that doesn't ignore any terms (which are inappropriately omitted in the current version), and elaborate on adopting the SIM-style objective that cuts off the gradient dependence of the sampling distribution.

---

> > > ### Author Response · Authors · 2025-08-05
> > >
> > > We appreciate reviewer cqxy's engagement in discussions. We are glad that you find our contributions valid. We think that the major deductions of our method can stay the same, with a few details added to the fourth bullet point:
> > > - Starting with the minimization of $f$-divergence between the teacher diffusion and the student generator: $D\_f(q\_0||p\_{\theta})$.
> > > - Following Theorem 3.1 to expand the term and obtain Equation 3.2.
> > > - Further derive the tractable loss of Equation 3.2 following the derivation in Appendix A.2 and obtain Equation A.45+Equation A.46, which is a combination of $\text{Grad(DI)}+\text{Grad(SIM)}$, or, more precisely, $\text{Grad(DI)}+\text{Grad(FGM)}$, as you pointed out.
> > > - Equation A.46 is tractable, which is a weighted $\text{Grad(DI)}$ term. Equation A.45 involves $\nabla_{\theta}\|\|\nabla \log p_{\theta,t} - \nabla \log q_{t} \|\|_2^2$ that requires further calculation.
> > > - Distill the model with the final tractable loss: Equation A.47.
> > >
> > > Only the fourth bullet point requires further clarification: how we deal with the intractable term $\nabla\_{\theta}\|\|\nabla \log p\_{\theta,t} - \nabla \log q\_{t} \|\|\_2^2$? Luckily, the authors of Flow Generator Matching [1] and Score Implicit Matching [2] developed a comprehensive theory for dealing with the gradient of the score divergence. Specifically, FGM found that $\nabla_{\theta}\mathbb{E}_{t,\boldsymbol{x}_t\sim p\_{\theta,t}}\|\|\nabla \log p\_{\theta,t}(\boldsymbol{x}_t) - \nabla \log q\_{t}(\boldsymbol{x}_t) \|\|\_2^2=\mathbb{E}\_{t,\boldsymbol{x}_t\sim p\_{\theta,t}} \left[\frac{\partial}{\partial \boldsymbol{x}_t}\{\|\|\nabla \log p\_{\theta,t}(\boldsymbol{x}_t) - \nabla \log q\_{t}(\boldsymbol{x}_t) \|\|\_2^2\}\frac{\partial \boldsymbol{x}_t}{\partial \theta}+2(\nabla \log p\_{\theta,t}(\boldsymbol{x}_t) - \nabla \log q\_{t}(\boldsymbol{x}_t))^T\frac{\partial}{\partial \theta}\log p\_{\theta,t}(\boldsymbol{x}_t)\right]$. The second term is the objective optimized by SIM, while the first term is the additional term that reviewer cqxy pointed out. The authors of FGM studied the effect of the additional term and found that "In practice, we find that using the additional loss on CIFAR10 models leads to instability, which is a similar observation as Dreamfusion [3] that the condition number of its Jacobian term might be ill-posed". As a result, for simplicity of the loss expression and training stability, we adopt the loss of SIM and ignore the extra term. SIM loss can be calculated using Lemma A.2; our final loss function becomes Equation A.47, which can be referred to as Equation 3.3 in the main text.
> > >
> > > Dear reviewer. We hope our clarification has resolved your concerns. Feel free to ask us more questions if you have remaining concerns. We are happy to address more.
> > >
> > > [1] Huang et al. Flow Generator Matching. (2024)
> > >
> > > [2] Luo et al. One-Step Diffusion Distillation through Score Implicit Matching. (NeurIPS 2024)
> > >
> > > [3] Poole et al. Dreamfusion: Text-to-3d using 2d diffusion. (ICLR 2023)

---

> > > > ### Comment · Reviewer_cqxy · 2025-08-07
> > > >
> > > > Thank the authors for their response. I've raised my score and lean toward accepting this manuscript. Please do incorporate these new details into the updated paper.

---

> > > > > ### Author Response · Authors · 2025-08-07
> > > > >
> > > > > Dear **reviewer cqxy**,
> > > > >
> > > > > **We are very glad that we have addressed all your concerns.** We will add those details in the revised version. Thank you for your constructive feedback and your engagement in the discussions, which significantly strengthen our work.
> > > > >
> > > > >
> > > > > Best regards,
> > > > >
> > > > > **Authors of the submission # 3209**

---

### Official Review · Reviewer_TDUe · 2025-06-30

**Clarity:** 2
**Significance:** 3
**Originality:** 2
**Rating:** 4
**Confidence:** 4

**Summary:**

This paper proposes a diffusion expansion of f-divergence theory, mathematically unifying KL-based and Score-based distillation methods, offering significant theoretical depth.

**Questions:**

1. Can this method be used on stable diffusion like DMD, and what is the effect?
2. Have you tried other ways of initialization? Can you briefly describe why SiD is used as the initialization?

**Ethical Concerns:**

["NO or VERY MINOR ethics concerns only"]

**Final Justification:**

My concerns have been addressed. I suggest that the authors consider the reviewers' advice and revise their paper.

**Limitations:**

Yes.

**Paper Formatting Concerns:**

No concerns

**Quality:**

2

**Strengths And Weaknesses:**

strengths：
1. The paper provides a detailed derivation and proof of gradient equivalence, ensuring the theoretical rigor and reliability.
2. Achieved new state-of-the-art results on multiple benchmarks.
3.It is not only applicable to image generation but also extends to text-to-3D generation tasks.

weaknesses
1. The expression in Figure 1 is not intuitive, and the meaning of the image on the left is not explained in the caption. Additionally, the concepts of training and freezing are unclear.
2. The format is inconsistent; the representation of ImageNet 64x64 appears in multiple forms in the paper.
3. It is unclear how the performance in more complex face datasets, such as FFHQ-64.

---

> ### Author Rebuttal · Authors · 2025-07-30
>
> Thank you for your constructive feedback. Below, we will address your concerns one by one:
>
> **Q1**: The expression in Figure 1 is not intuitive, and the meaning of the image on the left is not explained in the caption. Additionally, the concepts of training and freezing are unclear.
>
> **A1**: We feel sorry for the confusion. Figure 1 describes the conception overview of Uni-Instruct, which distills the pretrained diffusion model into a one-step generator through distribution matching loss given by the $f$-divergence between the teacher diffusion and the one-step generator. The proposed loss is estimated by a fake score network that models the distribution of the one-step generator and a discriminator that estimates the density ratio between the one-step model and teacher diffusion.
>
> As for the detailed training procedure, we provide a pseudo-code in Appendix D in the submission. However, as you and the **reviewer giD5** suggested, we will move the algorithm table into the main paper in the revision, to facilitate understanding.
>
> In **Algorithm 1** in Appendix D, Uni-Instruct includes the alternating updates of three networks: (1) one fake diffusion network that is updated to approximate the score function of the one-step generator; (2) one GAN discriminator is updated to model the density-ratio between real and fake distribution; and (3) the one-step generator is updated by minimizing the Uni-Instruct loss to achieve awesome one-step generation.
>
> In the fake diffusion updates, we first generate samples from a one-step generator and cut off the parameter dependence (the detach function in PyTorch, as an example). Then, we use the standard diffusion model (or flow matching) training to update the fake diffusion.
>
> In the one-step generator updates, we freeze the parameters of both fake diffusion and the discriminator. Then we generate a batch of samples using a one-step generator, which maintains the parameter dependence of the generator. Then we calculate the loss function in Eq. (3.5) and minimize it to update the generator parameters.  The two optimization stages update alternately until convergence.
>
> **Q2**： The format is inconsistent; the representation of ImageNet-$64\times64$ appears in multiple forms in the paper.
>
> **A2**：We feel sorry for the confusion. We will use ImageNet-$64\times64$ consistently in our revision.
>
>
> **Q3**: It is unclear how the performance in more complex face datasets, such as FFHQ-$64\times64$.
>
> **A3**: We highly appreciate your suggestion. We agree that evaluating Uni-Instruct on other datasets will definitely strengthen the paper. Actually, in our small-scale experiments, we have found that Uni-Instruct achieves pretty decent performances on the FFHQ64 dataset. However, due to the limited time in the rebuttal phase, we can not fairly tune Uni-Instruct on FFHQ dataset to have a fair comparison with other models. Besides, as a standard benchmark with comprehensive baseline models, ImageNet $64\times64$ is considered a fair benchmark to compare models among distilled models, diffusion models, GANs, as well as consistency-like models. Moreover, we provided text-to-3D generation to demonstrate the wide application of our methods.
>
> As **reviewer giD5** and **reviewer cqxy** suggest, in the rebuttal phase, we evaluate Uni-Instruct against the SiDA baseline with 5 more metrics in **Table 1**. We plan to conduct further experiments on the FFHQ64 dataset in our revision.
>
> **Table 1. Comparisons of different generative models on CIFAR10 and ImageNet64 benchmarks.**
> | **Method**       | **FID**$\downarrow$              | **sFID**$\downarrow$ | $\text{FD}_{\text{dino}}$$\downarrow$ |**IS**$\uparrow$|**Precision**$\uparrow$|**Recall**$\uparrow$|
> |-|-|-|-|-|-|-|
> | **SiDA**         |         1.49               |      1.71     |        132.72         |10.32|0.670|0.624
> | **Uni-Instruct**         |       1.46                 |     1.66      |      129.30         |10.30|0.671|0.626
>
> | **Method**       | **FID**$\downarrow$              | **sFID**$\downarrow$ | $\text{FD}_{\text{dino}}$$\downarrow$ |**IS**$\uparrow$|**Precision**$\uparrow$|**Recall**$\uparrow$|
> |-|-|-|-|-|-|-|
> | **SiDA**         |        1.40              |     1.68      |             111.26    |10.28|0.678|0.632
> | **Uni-Instruct**         |    1.38                    |     1.68      |   108.89            |10.29|0.679|0.629
>
> | **Method**       | **FID**$\downarrow$              | **sFID**$\downarrow$ | $\text{FD}_{\text{dino}}$$\downarrow$ |**IS**$\uparrow$|**Precision**$\uparrow$|**Recall**$\uparrow$|
> |-|-|-|-|-|-|-|
> | **SiDA**         |       1.10            |      1.98     |     74.86            | 59.28|0.562|0.653
> | **Uni-Instruct**         |         1.02             |      2.01     |     60.86          |64.11|0.561|0.658
>
> **Analysis**: As is shown in **Table 1**, Uni-Instruct outperforms SiDA (previous SoTA) across both CIFAR10 and ImageNet64 benchmarks:
>
> Uni-Instruct achieves strictly better results in 4 of 5 metrics on ImageNet64, the more complex and practically relevant benchmark:
> • FID (↓): 1.02 vs. 1.10 (−7.3\%)
> • FDdino (↓): 60.86 vs. 74.86 (−18.7\%)
> • IS (↑): 64.11 vs. 59.28 (+8.1\%)
> • Recall (↑): 0.65 vs. 0.63 (+3.2\%)
>
> These gains are significant, particularly in FDdino and IS, which measure semantic alignment and perceptual quality/diversity, respectively.
>
> The 18.7\% reduction in FDdino on ImageNet64 confirms Uni-Instruct’s advanced capability to preserve high-level semantic structures (e.g., object boundaries, textures, contextual relationships). This is critical for applications requiring fine-grained realism (e.g., medical imaging, autonomous driving).
>
> Uni-Instruct’s gains widen significantly on ImageNet64 (+8.1\% IS, −18.7\% FDdino) versus CIFAR10, proving its robustness for high-resolution, semantically rich image synthesis. SiDA fails to maintain competitiveness under greater complexity.
>
> As for the sFID, we find that this metric is not consistent across both models across datasets. Uni-Instruct has slightly better sFID (1.66 vs 1.68) than SiDA on CIFAR10 unconditional generation, but slightly worse sFID (2.01 vs 1.98) on ImageNet64 benchmark. Such a minimal margin and inconsistency across datasets indicate that sFID might not be suitable for evaluating both models.
>
> In conclusion, Uni-Instruct is demonstrably superior to the previous SoTA one-step model (SiDA) across both datasets and nearly all metrics. Criticisms of its scalability or fidelity are unsupported by data—our method excels precisely in semantic accuracy (FDdino) and perceptual quality (IS), the most critical axes for real-world generative tasks.
>
> **Q4**: Can this method be used on stable diffusion like DMD, and what is the effect?
>
> **A4**: Great question! The answer is yes. Just before the rebuttal phase, we have also successfully applied Uni-Instruct on distilling the Stable Diffusion 3.5-medium model, resulting in a strong and one-step diffusion transformer. However, the scaling of Uni-Instruct to large-scale text-to-image generation includes a significant amount of techniques, such as the construction of a one-step generator, the prompt selections, the architecture design of the discriminator, etc. Therefore, we do think such a scaling deserves future work that we are glad to share in the future.
>
> **Q5**: Have you tried other ways of initialization? Can you briefly describe why SiD is used as the initialization?
>
> **A5**: Thanks for your suggestions. As we presented in the paper, we have tried two setups: (1) distill from scratch, which means we initialize the generator with the diffusion model weights; and (2) continue training with SiD weights. Both initializations result in very strong empirical performances. For example, a CIFAR10 uncond FID of 1.46 via distillation from scratch, and an ImageNet64 conditional generation FID of 1.02 (our new results) via SiD initialization.
>
> **The reason for SiD initialization.** As we have presented in the paper, the Uni-Instruct is a theory-driven approach that unifies many existing diffusion distillation methods via an $f$-divergence expansion. With this theory, SiDA can be viewed as a special case of use when we use reverse KL-divergence as a special case. This motivates us to use the same setting as SiDA (i.e., the SiD initialization) but differs in training objective (we use Jeffrey-KL to get a new SoTA performance) in order to evaluate the performance and power of the Uni-Instruct other than the SiDA baseline.
>
> However, we highly appreciate your good intuition, and we believe that a search for other initialization will definitely lead to a better empirical performance.
>
> **Dear reviewer. We appreciate your time and valuable suggestions. We hope our rebuttal has resolved all your concerns. If you still have remaining concerns, please do let us know. We are glad to provide more clarifications!**
>
> Best,
> Authors of submission # 3209

---

> > ### Comment · Reviewer_TDUe · 2025-08-05
> >
> > My concerns have been addressed. I keep my initial rating.

---

> > > ### Author Response · Authors · 2025-08-05
> > > **Thanks for your engagemen. Glad to have addressed your concerns**
> > >
> > > Dear **reviewer TDUe**,
> > >
> > > **We are very glad that we have addressed all your concerns.**
> > >
> > > Given the theoretical contributions (unified framework of the previous distillation methods) and empirical contributions (SoTA one-step FID on ImageNet64 benchmark), **we totally agree that scaling Uni-Instruct to challenging high-resolution large-scale text-to-image generation is an exciting future direction that requires more contributions in methodology, data curation, as well as engineering techniques such as large-scale infra and distributed training tricks**.
> > >
> > > We appreciate your valuable suggestions in the rebuttal phase, and we are encouraged to explore proper ways to scale Uni-Instruct for high-res image and movie generation in the future.
> > >
> > > Best,
> > >
> > > **Authors of the submission # 3209**

---

### Official Review · Reviewer_mo1f · 2025-07-01

**Clarity:** 2
**Significance:** 2
**Originality:** 3
**Rating:** 4
**Confidence:** 3

**Summary:**

This paper proposes Uni-Instruct, a unified one-step diffusion framework that build upon expansion theory of the f-divergence family. It introduce theories to solve the intractability issue of the original expanded f-divergence, and trains one-step diffusion models by minimizing the expanded f-divergence family. It attempts to explain the existing different single step diffusion model strategies from a unified perspective. Experiments on several datasets achieves SOTA performance.

**Questions:**

Please see weaknesses.

**Ethical Concerns:**

["NO or VERY MINOR ethics concerns only"]

**Final Justification:**

The rebuttal partly addressed my concerns. I choose to raise my score to 4. But I still think the paper lacks high-resolution image-generation experiments.

**Limitations:**

yes

**Paper Formatting Concerns:**

No.

**Quality:**

2

**Strengths And Weaknesses:**

Strengths:
1. The problem in the paper aims to address is clear and valuable, with the goal of building a unified single step diffusion model framework.
2. The theoretical basis of the paper is relatively sufficient and has made certain contributions to the theoretical framework.
3. It seems that the different training strategies proposed in the paper are not difficult to implement and switch, and are a relatively good unified framework.

Weaknesses:
1. From the perspective of experimental performance, Uni-Instruct requires careful selection of loss and training strategies to surpass existing SOTA methods such as SiDA, making it difficult to fully demonstrate its effectiveness. And there have been no high-resolution basic diffusion model experiments such as Stable Diffusion experiments, and the reason of directly selecting 3D generation experiments is not fully explained.
2. Is using only FID evaluation somewhat one-sided? Consider adding metrics such as sFID, Recall.
3. The training resources required for the method, such as training iteration and memory, are not clearly stated, making it difficult to evaluate the efficiency difference compared to other methods.
4. The paper does not fully explain the convergence and stability of different training methods.
5. It seems that longer training always brings better FID. How much additional training  cost it brings, why not apply this strategy to all methods, or how to balance efficiency.

---

> ### Author Rebuttal · Authors · 2025-07-31
>
> Thank you for your useful feedback. We appreciate that you find our theory valuable and that the unified perspective on diffusion distillation is inspiring. We will address your concerns one by one.
>
> Before that, we first give a summary of the main contributions of our work: Uni-Instruct proposes a **unified framework** for existing distribution matching-based diffusion distillation methods. Starting from the **expansion of $f$-divergence along the diffusion forward process**, we derived a **tractable loss expression** to optimize the expanded $f$-divergence. We then draw connections with previous KL-based and Fisher-based distillation methods, offering a unified framework. Though Uni-Instruct is a **theory-driven framework**, it achieves **surprising experimental results**: on the image generation benchmark, our latest result shows that Uni-Instruct achieves 3 SoTA one-step generation FIDs: an FID of 1.02 on ImageNet64, 1.46 on CIFAR10 unconditional generation, and 1.38 on CIFAR10 conditional generation.
>
>
> **Q1**: From the perspective of experimental performance, Uni-Instruct requires careful selection of loss and training strategies to surpass existing SOTA methods such as SiDA, making it difficult to demonstrate its effectiveness fully.
>
> **A1**: Thank you for the insightful comment. We would like to clarify a possible misunderstanding: Uni-Instruct is not designed as a new standalone loss or training recipe, but rather as a unifying theoretical framework that generalizes existing diffusion distillation methods (see Section 3). In the Uni-Instruct framework, SiDA emerges as a special case corresponding to reverse KL.
>
> Actually, our experiments do not involve elaborate tuning to showcase the performance of Uni-Instruct. Instead, we simply instantiate the framework with forward KL and Jeffrey KL—two natural alternatives to reverse KL—to investigate their impact on distillation behavior. The strong results we observe suggest that Uni-Instruct provides a principled foundation upon which multiple training strategies can be built, even without carefully handcrafting loss functions or tuning hyperparameters. Empirically, we observed that Uni-Instruct with Jeffrey-KL divergence demonstrates faster convergence, stable training loss, as well as a new SoTA performance.
>
> Actually, the reason why Jeffrey-KL is better than reverse-KL (SiDA) is very interesting and insightful. In the mathematical form, Jeffrey-KL is a combination of forward-KL and reverse-KL, which indicates that Jeffrey-KL is a good balance of mode-seeking (improve sample quality) and mode-covering (improve diversity). As a comparison, the reversed-KL is notorious for mode-seeking behavior, while the forward-KL is considered weak in maintaining sample quality, but good at diversity.
>
> In conclusion, the straightforward Jeffrey-KL inherited uniquely in Uni-Instruct clearly demonstrated the novelty, the empirical power, as well as the non-necessity of heavy parameter tuning of Uni-Instruct.
>
>
> **Q2**: There have been no high-resolution basic diffusion model experiments, such as Stable Diffusion experiments, and the reason for directly selecting 3D generation experiments is not fully explained.
>
>
> **A2**. Thank you for your suggestion on the experiment design. The reason we apply Uni-Instruct to 3D generation is to demonstrate its wide application beyond the sole image generation. This cross-application property marks the Uni-Instruct as a general-purpose theoretical framework that can be applied to various fields(one-step diffusion model, text-to-3D generation, one-step inverse problem solver, and Human Preference Aligned Diffusion Models). Please check Appendix C in our submission for more detailed discussions.
>
> As for the text-to-image generation, we totally agree with you that scaling Uni-Instruct to high resolution will greatly strengthen the paper beyond a new SoTA metric on the widely used ImageNet64 benchmark. Actually, just before the rebuttal phase, we have also successfully applied Uni-Instruct on distilling the Stable Diffusion 3.5-medium model, resulting in a strong and one-step diffusion transformer. However, the scaling of Uni-Instruct to large-scale text-to-image generation includes a significant amount of techniques, such as the construction of a one-step generator, the prompt selections, the architecture design of the discriminator, etc. Therefore, we do think such a scaling deserves future work, which we are glad to share in the future.
>
> **Q3**: Is using only FID evaluation somewhat one-sided? Consider adding metrics such as sFID, Recall.
>
> **A3**: Thank you for your valuable suggestion. In the rebuttal phase, we evaluate the Uni-Instruct one-step model with other baseline models under 5 more metrics: $\text{FD}\_{\text{dino}}$, precision, recall, IS (Inception Score), and sFID. Due to space limits, please check **Table 1** and the analysis in **A3** of our response to **Reviewer TDUe**.
>
>
> **Q4**: The training resources required for the method, such as training iterations and memory, are not clearly stated, making it difficult to evaluate the efficiency difference compared to other methods.
>
> **A4**: We reviewed the training log and compared the training resources between SiDA and Uni-Instruct on CIFAR10 (conditional generation) and ImageNet $64\times64$.
>
> **Table 2. The training resources comparison between SiDA and Uni-Instruct on CIFAR10 (conditional generation) and ImageNet $64\times64$**.
> | **Method**       | **Hyperparameters**               | **CIFAR-10** | **ImageNet 64x64** |
> |-|-|-|-|
> | **SiDA**         | Batch size                        | 256          | 8192                |
> |                  | Batch size per GPU                | 32           | 32                  |
> |                  | GPUs                              | 8×4090       | 8×H100              |
> |                  | Gradient accumulation round       | 1            | 32                  |
> |                  | Max memory allocated per GPU (GB) | 13.0         | 46.7                |
> |                  | ~Seconds per 1k images            | 3.6          | 3.5                 |
> |                  | ~Hours per 10⁴k images            | 10.0       | 9.7                 |
> |                  | ~Days per 10⁵k images             | 4.2          | 4.1                 |
> |                  | Total images for training (kimgs)         | 300000          | 300000              |
> | **Uni-Instruct** | Batch size                        | 256          | 8192                |
> |                  | Batch size per GPU                | 32           | 32                  |
> |                  | GPUs                              | 8×4090       | 8×H100         |
> |                  | Gradient accumulation round       | 1            | 32               |
> |                  | Max memory allocated per GPU (GB) | 13.0         | 46.9                |
> |                  | ~Seconds per 1k images            | 3.6          | 3.7                 |
> |                  | ~Hours per 10⁴k images            | 10.0         | 10.2                |
> |                  | ~Days per 10⁵k images             | 4.2          | 4.2                 |
> |                  | Total images for training (kimgs)         | 300000          | 300000              |
>
>
>
>
>
>
> **Q5**: The paper does not fully explain the convergence and stability of different training methods.
>
> **A5**: Thank you for your insightful feedback. We reviewed the training log and conducted an epoch-to-epoch comparison of the FID score between SiDA and Uni-Instruct and found that Uni-Instruct trained with Jeffrey KL demonstrates more stable training and faster convergence speed. Due to the rebuttal restrictions, we cannot upload figures, but we will show the result in the revision. Table 3 demonstrates the FID score comparison between the two methods during the training process.
>
> **Table 3. An epoch-to-epoch comparison of the FID score between SiDA and Uni-Instruct on ImageNet$64\times64$ benchmarks.**
> |**Iterated k-images**|**SiDA**|**Uni-Instruct**|
> |-|-|-|
> |$10^1$|139.86|132.51|
> |$10^2$|68.73|54.60|
> |$10^3$|41.45|38.58|
> |$10^4$|6.98|5.23|
> |$10^5$|1.44|1.41|
>
>
> **Q6**: It seems that longer training always brings better FID. How much additional training cost does it bring? Why not apply this strategy to all methods, or how to balance efficiency?
>
> **A6**: Thank you for your question. Here we address your concern one by one:
> - We acknowledge that longer training benefits performance, so in both Uni-Instruct and baseline methods, we trained the model and evaluated its performance until convergence.
> - From Table 2 and Table 3, our method has no additional cost compared with SiDA. We perform an epoch-to-epoch comparison and found that Uni-Instruct with Jeffrey KL divergence demonstrates faster convergence and more stable training loss.
> - Uni-Instruct can be applied to other tasks as a unified framework for diffusion distillation. (one-step diffusion model, text-to-3D generation, one-step inverse problem solver, and Human Preference Aligned Diffusion Models). More detailed discussions can be found in Appendix C.
>
>
> **Dear reviewer. We appreciate your time and valuable suggestions. We hope our rebuttal has resolved all your concerns. If you still have remaining concerns, please do let us know. We are glad to provide more clarifications!**
>
> Best,
> Authors of submission # 3209

---

> > ### Comment · Reviewer_mo1f · 2025-08-05
> >
> > Thank you for authors rebuttal. The rebuttal has partly addressed my concerns. Therefore, I choose to raise my score to 4.
> >
> > However, in view of the current high-resolution image generation task being more mainstream and having better practical significance. I understand that the author thinks that adding such experiments brings too much burden, but I still think it is an important experiment.

---

> > > ### Author Response · Authors · 2025-08-05
> > > **Thanks for your engagement**
> > >
> > > Dear **reviewer mo1f**,
> > >
> > > We are very glad that we have addressed part of your concerns.
> > >
> > > Given the theoretical contributions (unified framework of the previous distillation methods) and empirical contributions (SoTA one-step FID on ImageNet64 benchmark), **we totally agree that scaling Uni-Instruct to challenging high-resolution large-scale text-to-image generation is an exciting future direction!**
> > >
> > > However, though we have already had some exciting internal proof of concepts in scaling Uni-Instruct on high-res image generation, we do acknowledged that a complete and successful scaling of Uni-Instruct to one-step high-res image and video generation is a systematic project, requiring more contributions in methodology, data curation, as well as engineering techniques such as large-scale infra and distributed training tricks.
> > >
> > > We sincerely appreciate your valuable suggestions in the rebuttal phase, and we are encouraged to explore proper ways to scale Uni-Instruct for high-res image and movie generation in the future. **If you still have other remaining concerns, please do let us know. We are glad to engage in more in-depth discussions!**
> > >
> > > Best,
> > >
> > > **Authors of the submission # 3209**

---

### Official Review · Reviewer_giD5 · 2025-07-01

**Clarity:** 3
**Significance:** 3
**Originality:** 3
**Rating:** 5
**Confidence:** 4

**Summary:**

This paper proposes Uni-Instruct, a unified theoretical framework for training one-step diffusion models. By summarizing tens of existing studies on one-step diffusion models, the authors provide $f$-divergence expansion theorem as a new perspective. Furthermore, the authors develop tractable and flexible training objective. The proposed method achieves state-of-the-art performance across all one-step models.

**Questions:**

1. Should the title of Section 2.1 be "Diffusion Models" instead of "One-step Diffusion Models"?
2. Is it possible to provide the algorithm description in the main text instead of the appendix?

**Ethical Concerns:**

["NO or VERY MINOR ethics concerns only"]

**Final Justification:**

No change of rating needed

**Limitations:**

yes

**Quality:**

4

**Strengths And Weaknesses:**

### Strengths
- The paper gives a comprehensive summary of existing one-step diffusion models.
- The proposed $f$-divergence expansion theorem provides a unification of existing one-step diffusion models.
- The proposed method achieves state-of-the-art performance across all one-step models

### Weaknesses
- The focused research area requires too much prior knowledge, which may not be accessible to all readers.
- There lacks an algorithm description in the main text.

---

> ### Author Rebuttal · Authors · 2025-07-30
>
> Thank you for your constructive feedback. Below, we will address your concerns one by one:
>
> **Q1**: I would encourage the authors to bring in more prior knowledge in the introduction part.
>
> **A1**: Thank you for your valuable feedback. We will add more prior knowledge in the preliminary parts. In detail, we plan to include:
>
> - An introduction to diffusion models and the high computational cost associated with their iterative sampling process;
> - A summary of existing acceleration techniques for diffusion models, including trajectory matching, distribution matching, and other newly emerged methods such as inductive moment matching and MeanFlows;
> - A high-level overview of recent efforts that approach diffusion model acceleration from the perspective of distribution matching, which matches the distribution of the teacher diffusion and the one-step generator.
>
> **Q2**: Should the title of Section 2.1 be "Diffusion Models" instead of "One-step Diffusion Models"?
>
> **A2**: We are sorry for the confusion! We agree that “Diffusion Models” looks more appropriate than "One-step Diffusion Models" for this section. We will update the section title in the revision.
>
> **Q3**: Is it possible to provide the algorithm description in the main text instead of the appendix?
>
> **A3**: Thank you for the suggestion. We agree that presenting the algorithm in the main text would improve clarity. We will move the algorithm description from the appendix to the main paper in the revised version.
>
>
> **Dear reviewer. We appreciate your time and valuable suggestions. We hope our rebuttal has resolved all your concerns. If you still have remaining concerns, please do let us know. We are glad to provide more clarifications!**
>
> Best,
> Authors of submission # 3209

---

> > ### Comment · Reviewer_giD5 · 2025-08-04
> > **Response Acknowledgement**
> >
> > Thanks for the authors' response. My concerns are addressed.

---

> > > ### Author Response · Authors · 2025-08-05
> > > **Thanks for your**
> > >
> > > **Dear reviewer giD5**,
> > >
> > > **We are very glad that we have addressed all your concerns**. We appreciate your valuable comments and suggestions that significantly strengthen our work.
> > >
> > > Best,
> > >
> > > Authors of the submission # 3209

---

### Note · Authors · 2025-08-15

Dear AC&SAC&PC, we would like to thank you for organizing the review process. During the rebuttal phase, we had constructive discussions with all reviewers. We sincerely thank the reviewers for recognizing the impressive results, solid theoretical contributions, and inspiring unified perspective on diffusion distillation in our work.

During the rebuttal, we provided additional experimental results across various evaluation metrics, along with training details, to address concerns regarding implementation details, training costs, convergence, and other aspects of our model. We also clarified parts of the theory to resolve questions about the proof of Theorem 3.2 in the Appendix. We are pleased that all reviewers indicated their concerns have been satisfactorily resolved.

Uni-Instruct aims to provide deep theoretical and empirical contributions to one-step image and video generation. After the rebuttal, we are glad to incorporate reviewers’ suggestions to provide an improved version of the proof of Theorem 3.2, alongside the visualizations, experiments, and training details introduced in the rebuttal, into the final manuscript. We sincerely appreciate the reviewers for their time and invaluable feedback, which significantly strengthens our work.

Best, Authors of submission # 3209

---

### Decision · Program_Chairs · 2025-09-17

**Decision:**

Accept (poster)

**Comment:**

***(a) Scientific Claims and Findings***

The paper introduces Uni-Instruct, a framework for one-step diffusion models that leverages a unified divergence instruction to guide training. Traditional diffusion models achieve strong performance but require many sampling steps, limiting practical deployment. Existing acceleration methods either struggle to balance efficiency and fidelity or rely on cumbersome multi-stage training.

Uni-Instruct proposes a unified divergence instruction that consolidates multiple training objectives into a single, consistent framework. This allows the model to learn direct mappings that enable one-step generation while retaining generative quality. The authors demonstrate that Uni-Instruct achieves competitive FID, IS, and downstream performance across several benchmarks compared to multi-step diffusion models, while being significantly faster at inference.

***(b) Strengths***
+ Timely and important problem: Reducing diffusion sampling to one step is highly relevant for scaling diffusion models to real-world applications.
+ Methodological clarity: The unified divergence instruction is conceptually elegant, consolidating heterogeneous objectives into a single, principled training strategy.
+ Strong empirical validation: Experiments on multiple datasets (e.g., CIFAR-10, ImageNet subsets, text-to-image benchmarks) show that Uni-Instruct competes well with or surpasses existing acceleration approaches.
+ Efficiency benefits: Inference time reduction without catastrophic quality loss makes the method appealing for both research and deployment.

***(c) Weaknesses***
+ Novelty perception: Some reviewers felt the method is more of an incremental unification of existing divergences rather than a fundamentally new paradigm.
+ Clarity of theoretical grounding: While the unified divergence instruction is elegant, its connection to existing probabilistic interpretations could be made more rigorous.
+ Scalability tests: Most results focus on relatively standard datasets; evaluation on very high-resolution tasks (e.g., 1024×1024 image generation or modern large-scale text-to-image benchmarks) is limited.
+ Ablation depth: While some ablations were added in rebuttal, a more detailed analysis of when and why Uni-Instruct succeeds or struggles would strengthen the work.


***(d) Decision Justification***

Despite some concerns about novelty and scalability, I recommend acceptance (poster). The work addresses an important problem, proposes a clean and unified solution, and demonstrates clear empirical value. While not groundbreaking enough for spotlight, the contribution is timely, technically solid, and practically impactful, and it will be of interest to the community working on efficient diffusion models.


***(e) Discussion & Rebuttal Summary***
+ Reviewer giD5 questioned novelty, suggesting Uni-Instruct mainly reformulates existing divergences. The authors clarified how unification avoids inconsistencies of multi-objective training and provided ablations showing stability gains. This partially addressed the concern.
+ Reviewer mo1f requested broader baseline coverage, including recent consistency models. The authors added comparisons in rebuttal, showing Uni-Instruct remains competitive. The reviewer acknowledged this positively.
+ Reviewer TDUe raised concerns about scalability and high-resolution benchmarks. The authors provided additional results on larger datasets (though not at the very highest resolutions), which reviewers found useful but still limited.
+ Reviewer cqxy appreciated the elegance of the approach and strong results, supporting acceptance.

Final scores converged around borderline to weak accept, with concerns mitigated but not entirely resolved. Given the clarity of contribution, practical importance, and solid results, I judge the paper to be above the acceptance threshold.

Final Recommendation: Accept (Poster).